# MaRVL-QA: A Benchmark for Mathematical Reasoning over Visual Landscapes

## Abstract

A key frontier for Multimodal Large Language Models (MLLMs) is the ability to move beyond semantic description and perform structured spatial analysis directly from images. Mathematical surface plots provide a rigorous testbed for this capability, as they isolate systematic visual reasoning from the semantic noise of natural images. To measure progress on this frontier, we introduce **MaRVL-QA** (**Ma**thematical **R**easoning over **V**isual **L**andscapes), a new benchmark designed to quantitatively evaluate these foundational skills. The benchmark comprises two novel tasks: **Topological Counting**, which requires models to identify and enumerate local extrema; and **Transformation Recognition**, which tests their ability to detect applied geometric transformations. Generated from a curated library of functions with rigorous ambiguity filtering, our evaluation on MaRVL-QA reveals that even state-of-the-art MLLMs struggle significantly, often resorting to superficial heuristics instead of robust strategies. We present MaRVL-QA as a challenging diagnostic tool to expose current limitations and to guide the development of MLLMs with stronger and more systematic visual-mathematical abilities.

## 1 Introduction

The fusion of large language models with visual data has unlocked powerful new capabilities in artificial intelligence. These Multimodal Large Language Models (MLLMs) can interpret the visual world with remarkable fluency, moving far beyond simple object labeling. They excel at generating rich, detailed descriptions for complex scenes, answering nuanced questions about the relationships and interactions between objects, and even engaging in multi-turn, contextual dialogue about what they see OpenAI et al. (2024); Comanici et al. (2025); Liu et al. (2023b). At its core, this success comes from their ability to create a strong correspondence between natural language and the high-level semantic content of an image Radford et al. (2021); Li et al. (2023a).

However, this strength with high-level semantics stands in stark contrast to a fundamental limitation: a difficulty with precise spatial and structural analysis. This deficit is readily observed in natural image contexts, where these models often fail to accurately count objects, determine specific positional relationships, or interpret complex spatial arrangements Fu et al. (2024); Li et al. (2023b); Zhang et al. (2018); Grover et al. (2025); Tamarapalli et al. (2025). Diagnosing these failures in natural scenes is inherently challenging, since rich semantic context can mask whether a model is struggling with perception, reasoning, or both Johnson et al. (2017). Mathematical visualizations offer a more controlled alternative: they strip away semantic cues and isolate the task of interpreting structural and geometric properties. In this work, we leverage such plots to evaluate the foundational visual-mathematical skills that underpin more complex reasoning.

Concurrently, mathematical reasoning has been established as a key evaluation axis for language models, with benchmarks such as GSM8K Cobbe et al. (2021) and MATH Hendrycks et al. (2021) driving progress on complex, text-based quantitative problems. However, these benchmarks focus on reasoning from equations and word problems, leaving open the question of how well models can interpret mathematical concepts presented visually Lu et al. (2024). Tasks such as inferring properties of a function from its plot require interpreting topological and geometric structure in a semantically sparse setting - a capability that is not captured by existing evaluations

To address this gap, we introduce **MaRVL-QA: Mathematical Reasoning over Visual Landscapes**, a benchmark for testing how MLLMs connect visual perception with basic mathematical

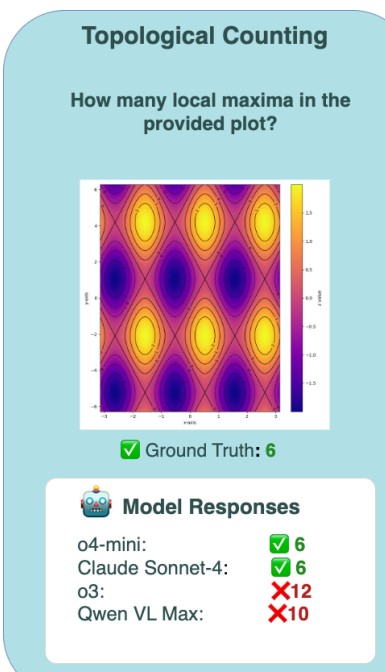
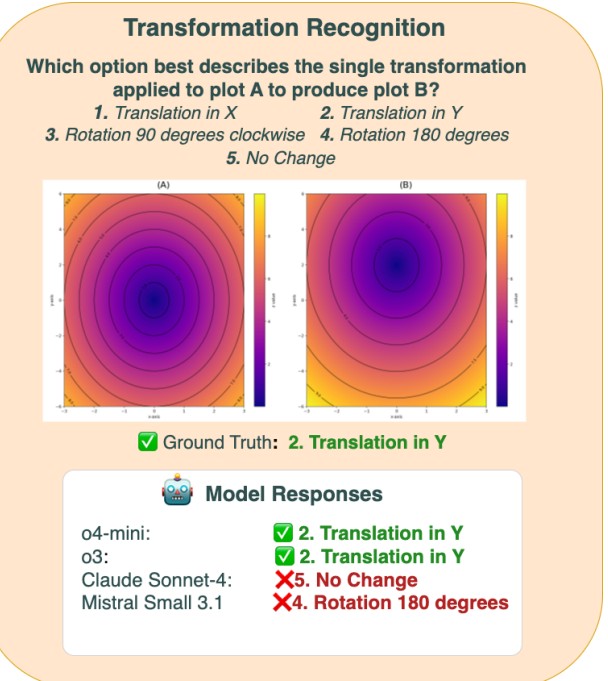

Figure 1: Illustration of the core MaRVL-QA tasks, showcasing the visual and textual prompts presented to the models. **(a) Topological Counting:** A single plot is shown, and the model must answer a direct question about the number of a specific topological feature, such as local maxima. **(b) Transformation Recognition:** An original plot and its transformed version are presented, and the model must select the correct transformation from a list of choices.

structure. It is designed as a diagnostic framework: it focuses on controlled tasks that isolate foundational visual-mathematical skills. It comprises two task categories, which represent essential building blocks for more advanced reasoning: **Topological Counting**, which assesses the identification of structural features; and **Transformation Recognition**, which tests their ability to detect applied geometric transformations.

This work makes the following primary contributions:

- We introduce *MaRVL-QA*, a two-task evaluation of intrinsic visual-mathematical skills (i) extrema counting and (ii)transformation recognition.
- We build the benchmark from a curated library of mathematical functions, programmatically generating more than 80,000 QA pairs with multi-stage ambiguity filtering to yield objective ground truth.
- We release *MaRVL-QA-Mini*, a 2,748-item, stratified subset balanced across styles and transformations for efficient, comparable evaluation.
- We evaluate modern MLLMs on MaRVL-QA, exposing systematic failure modes and gaps that are obscured by semantics in natural images.

## 2 RELATED WORK

### 2.1 CHART AND PLOT COMPREHENSION

A primary focus of multimodal research is reasoning over visual data representations. Foundational VQA benchmarks target natural images Antol et al. (2015); Goyal et al. (2017); Yerramilli et al. (2025). Plot-centric work evaluates data visualizations: early FigureQA and PlotQA test basic point/value retrieval Kahou et al. (2018); Methani et al. (2020); later ChartQA and related tasks increase semantic complexity and rely on text–visual fusion Masry et al. (2022); Hossain et al. (2022);

chart-to-table aims to reconstruct underlying data Liu et al. (2023a). Unlike these, MaRVL-QA is geometry-first and OCR-proof: axes/labels are held constant across pairs, so models must reason about the surface itself rather than read values; each validated item is rendered across diverse styles (color maps, contour/heatmap settings) to test style robustness controls that typical chart QA lacks.

## 2.2 EVALUATION OF SPATIAL REASONING

Spatial reasoning datasets document MLLM limitations Wu et al. (2023). CLEVR and successors benchmark *extrinsic* relations among discrete objects in synthetic or more realistic scenes, including direction distinctions Johnson et al. (2017); Liu et al. (2023c). MaRVL-QA instead targets *intrinsic* reasoning on a single continuous surface:(i) counting visually unambiguous extrema (topological primitives), and (ii) deciding which operation (rotation/translation) maps one rendering to another - providing a controlled testbed for claimed spatial equivariance.

## 2.3 MATHEMATICAL REASONING BENCHMARKS

Text-centric math benchmarks (e.g., GSM8K, MATH) emphasize problem solving from language Cobbe et al. (2021); Hendrycks et al. (2021); even when images appear (e.g., UniGeo), the task is primarily defined by text Chen et al. (2023). MaRVL-QA inverts this dependency: the visualization defines the problem. It measures grounding of mathematical concepts (topology, global transformations) directly in visual fields - capabilities not isolated by existing multimodal math benchmarks.

## 3 BENCHMARK GENERATION PIPELINE

The MaRVL-QA benchmark is systematically generated from a comprehensive library of three-dimensional functions, $z = f(x, y)$. To ensure mathematical diversity and analytical rigor, we explicitly hand-selected these function families rather than relying on random procedural generation, guaranteeing that every task is grounded in an objective, provable truth. We further refined this library by individually selecting the plotting domain for each function, ensuring its most significant visual features are centered and prominent. These curated functions are then programmatically rendered into plots and assembled into tasks, as detailed in the following sections. A full specification of the function families is available in the appendix A.1.

### 3.1 PLOT RENDERING

Each curated function is programmatically rendered into a high-resolution image from a densely sampled data grid. To test model robustness against superficial visual features, our rendering pipeline generates plots in several distinct styles. This is achieved by systematically varying both the plot type (e.g., heatmaps, contour plots) and cosmetic properties like color maps.

A critical design choice is to render all plots with their corresponding axes and numerical labels. This methodology compels a model to synthesize the visual information of the plotted surface with the symbolic information of the coordinate system, moving beyond simple shape recognition. As detailed in the following sections, additional measures are taken during the generation of each specific task to ensure the problems demand genuine spatial reasoning rather than simple text extraction. A full specification of the rendering parameters, plot styles, and color maps is provided in the appendix.

### 3.2 TASK SPECIFIC PIPELINES

#### 3.2.1 TOPOLOGICAL COUNTING

The Topological Counting task is designed to directly evaluate a model's ability to identify and enumerate fundamental topological features of a surface. The task focuses specifically on counting local maxima and minima, as these features manifest as discrete and perceptually unambiguous visual signatures (e.g., bright peaks and dark valleys). In contrast, features such as saddle points were excluded; they lack a single, clear visual signature in 2D renderings and can be indistinguishable from other features (like two nearby peaks), which undermines the goal of objective, reproducible ground truth.

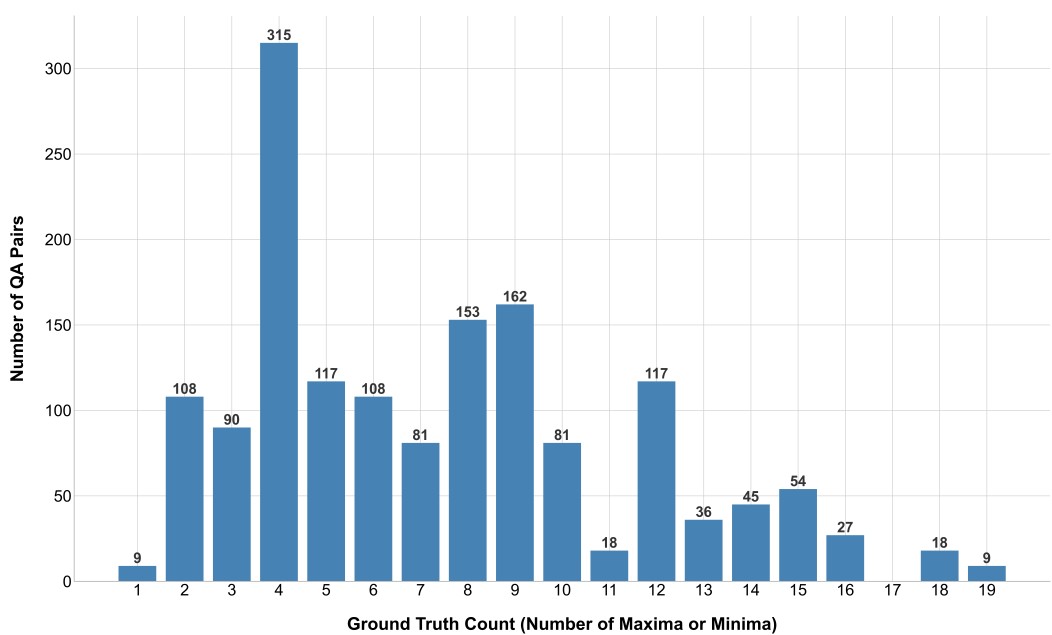

Figure 2: Distribution of correct answers in the 1548-item Topological Counting task.

The task is framed as a question-answering problem where the model is presented with a single image and asked a question such as, "How many local maxima are visible in this plot?" The generation of ground truth for this task is a multi-stage process designed to ensure precision and eliminate ambiguity, combining automated analysis with meticulous manual curation. The automated stage uses a hybrid coarse-to-fine strategy: an image-based algorithm finds approximate extrema locations on a dense data grid, which are then refined with a high-precision numerical optimization routine on the underlying continuous function.

Crucially, this is followed by a manual curation stage to certify that every question is fair and visually unambiguous. During this stage, authors reviewed every generated plot to filter out instances with ambiguous features (e.g., those near the plot boundary) and certified that the programmatically identified count exactly matched the number of clearly visible features. This hybrid pipeline yielded 1,548 high-confidence question-answering pairs, with a diverse distribution of ground-truth counts as illustrated in Figure 2. The full technical specification of the algorithms and review protocol is detailed in the appendix A.3.

### 3.2.2 TRANSFORMATION RECOGNITION

The Transformation Recognition task evaluates a model's holistic spatial reasoning by testing its ability to identify global geometric transformations. The task is structured as a multiple-choice problem and deliberately focuses on atomic transformations - single-axis translations and rotations because they admit a unique, unambiguous mapping between plots. More complex operations were avoided as they often result in visually indistinguishable cases, which complicates objective ground truth.

A core design principle is to compel genuine spatial reasoning rather than allowing solutions based on simple heuristics like Optical Character Recognition (OCR). To prevent models from inferring transformations by reading axis labels, we employ specific rendering strategies. For rotations, both plots are rendered within a unified, expanded coordinate domain; for translations, the surface is shifted within a fixed domain window. In both cases, the axis labels remain identical, forcing the model to reason about the surface's geometric features.

To guarantee that each question has a single, unambiguous visual answer, we implemented a rigorous, multi-stage filtering pipeline. This process programmatically eliminates two key sources of ambiguity: (1) symmetric transformations that result in no discernible visual change, and (2) con-

founding transformations where one operation is perceptually indistinguishable from another (e.g., a rotation appearing as a translation).

This crucial two-way validation ensures, for example, that an accepted translation is visually distinct from any possible rotation, and vice-versa. This pipeline produced 79,542 high-confidence QA pairs distributed across four transformation types. The full technical specification of our filtering algorithms and similarity metrics is detailed in the appendix.

## 3.3 THE MaRVL-QA-MINI TEST SET

To provide a focused and computationally tractable standard for evaluation, we constructed the MaRVL-QA-Mini test set. This set is composed of the complete, manually-curated Topological Counting task (1,548 QA pairs) and a high-quality, stratified subset of 1,200 QA pairs from the Transformation Recognition task. This sample size was chosen to provide sufficient statistical power for fine-grained analysis while ensuring the benchmark remains efficient for repeated model evaluation.

To create the Transformation Recognition subset and prevent confounding biases, we used a stratified sampling strategy. First, the subset was balanced by visual style consistency, with 600 pairs where the original and transformed plots share the same visual style and 600 pairs where their styles differ. Within each of these two groups, the data was further balanced by transformation type, with exactly 150 examples for each of the four transformation types (90-degree rotation, 180-degree rotation, x-translation, and y-translation). Finally, within these smaller blocks, a round-robin sampling strategy ensured a uniform distribution across the underlying function families, to the extent possible given our ambiguity filters. This principled construction makes MaRVL-QA-Mini a robust and reliable tool for model evaluation.

## 4 RESULTS AND ANALYSIS

We evaluated the performance of ten distinct Multimodal Large Language Models (MLLMs) on the Topological Counting and Transformation Recognition tasks. Our evaluation aimed to be as deterministic as possible (temperature = 0), and our system prompts (A.7) requested that models provide a direct answer in a specific format.

However, we observed that many models, particularly open-source variants, failed to consistently adhere to the requested format. For instance, some models would output the word "twelve" instead of the integer "12" for counting tasks. Because these inconsistencies render rule-based parsers brittle and prone to error, we employed a capable LLM (GPT-4.1) to interpret the model's intended answer from its full text output, ensuring fair and accurate evaluation across all systems (results without LLM parser in A.6.1).

The specific format of the ground truth was tailored to each task. For the Topological Counting task, the ground truth was an integer value representing the total count of maxima or minima. In the case of the Transformation Recognition task, presented as a five-option multiple-choice question (MCQ) labeled 1 through 5, the ground truth was the integer of the correct option.

## 4.1 TOPOLOGICAL COUNTING

We evaluate performance on the Topological Counting task using three metrics: Accuracy, and Relaxed Accuracy at 10% and 20% tolerance, which credits near-miss answers. The full results, including 95% confidence intervals, are in Table 1.

The results reveal a profound deficit in this capability across all tested models. The accuracy for the top-performing model, o4-mini, is only 58.91%, underscoring the fundamental difficulty of this task for current systems. However, the story is in the errors. o4-mini's accuracy jumps by nearly 17 points to 75.78% with a 20% tolerance, indicating that a significant portion of its mistakes are 'near-misses'.

This pattern does not hold for low-performing models. For LLaVA-13B, relaxing the accuracy criteria provides almost no benefit, with its score barely moving from 6.65% to 9.82%. This suggests its failures are not minor miscounts but catastrophic breakdowns in reasoning; when it is wrong, it

is not even close. This clear divergence where top models are often approximately correct while weaker models fail completely, points to the absence of a robust, generalizable counting mechanism in most MLLMs.

Table 1: Overall performance on the Topological Counting task. We report Strict Accuracy and Relaxed Accuracy at 10% and 20% tolerance, with 95% confidence intervals. The data highlights a key divergence in failure modes: top-performing models often make near-miss errors, while lower-performing models fail catastrophically.

| Model | Accuracy (%) | Relaxed Acc. (10%) | Relaxed Acc. (20%) |
|---|---|---|---|
| OpenAI o4-mini OpenAI (2025) | 58.91 ± 2.45 | 63.24 ± 2.40 | 75.78 ± 2.13 |
| Claude Sonnet 4 Anthropic (2025) | 53.94 ± 2.48 | 57.69 ± 2.46 | 66.28 ± 2.35 |
| OpenAI o3 OpenAI (2025) | 50.52 ± 2.49 | 55.30 ± 2.47 | 65.63 ± 2.36 |
| Qwen-2.5 VL Max Bai et al. (2025) | 41.80 ± 2.45 | 43.09 ± 2.46 | 56.46 ± 2.47 |
| Pixtral Large Mistral AI et al. (2024) | 38.37 ± 2.42 | 40.63 ± 2.44 | 55.43 ± 2.47 |
| Mistral 3.1 Small Mistral AI (2025b) | 38.18 ± 2.42 | 40.70 ± 2.44 | 54.33 ± 2.48 |
| Mistral 3.1 Medium Mistral AI (2025a) | 34.88 ± 2.37 | 36.56 ± 2.40 | 52.20 ± 2.49 |
| Llava 1.6 13B Liu et al. (2024) | 6.65 ± 1.24 | 6.65 ± 1.24 | 9.82 ± 1.48 |
| Llava 1.6 7B Liu et al. (2024) | 0.65 ± 0.42 | 0.65 ± 0.42 | 0.65 ± 0.42 |
| Llava 1.6 34B Liu et al. (2024) | 0.19 ± 0.25 | 0.19 ± 0.25 | 0.26 ± 0.28 |

### 4.1.1 ANALYSIS BY FEATURE COUNT

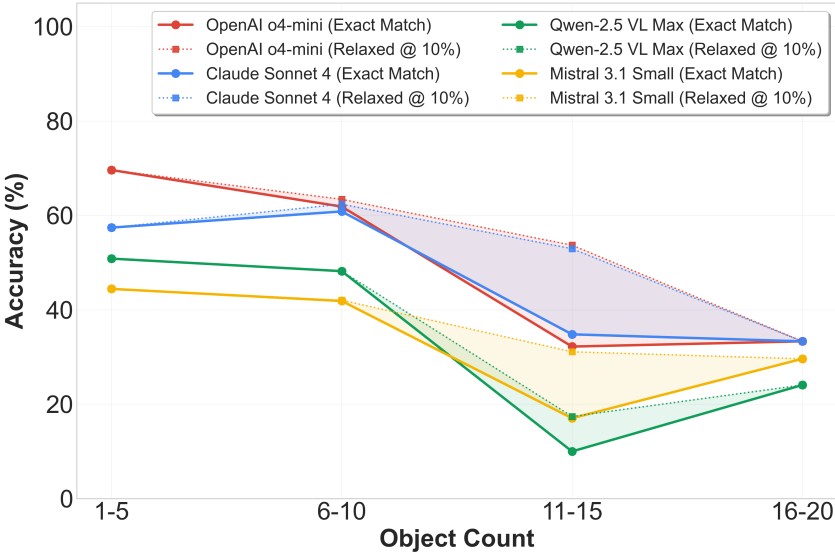

Figure 3: Model performance on Topological Counting as a function of the number of features to be counted. The plot shows the strict accuracy for four representative models, illustrating a sharp decline in performance as scene complexity increases. This demonstrates a critical scalability failure in current MLLMs. Error bands denote 95% confidence intervals.

The most critical factor impacting performance is the number of features to be counted. As illustrated in Figure 3, all competent models exhibit a sharp decline in accuracy as the scene complexity increases, a finding consistent with literature noting the brittleness of MLLM counting abilities. The full data is in Appendix Table 10.

**Small Counts (1-5 features):** In the range analogous to human "subitizing," models perform best. Even here, however, the task is not solved; the top-performing o4-mini achieves just 69.64% strict accuracy. **Moderate and High Counts (6+ features):** Performance collapses once the count exceeds the subitizing range. o4-mini's accuracy falls to 61.88% for the 6-10 bin and plummets to

just 32.22% for counts of 11-15. This steep decline suggests a fundamental failure in procedural enumeration - the process of systematically identifying and tallying individual items.

Interestingly, the relaxed accuracy metrics reveal a key nuance in this failure. In the 11-15 count range, while o4-mini's accuracy is only 32.22%, its accuracy at 20% tolerance is 71.11%. This widening gap at higher counts suggests that while models lose precision, they often retain an approximate sense of quantity. This points to a specific breakdown in the enumeration process rather than a complete loss of numerical understanding; the model seems to know there are "a lot" of features but loses the exact count.

### 4.1.2 ANALYSIS BY COUNT TYPE

A universal bias is observed across all models: they are consistently less accurate at counting minima than maxima. This performance gap is not due to task difficulty; a weighted average calculated from our ground-truth frequencies shows that maxima-counting tasks require counting slightly more items (a mean of 7.53) than minima tasks (a mean of 7.15). This suggests the gap stems from the lower visual salience of minima (dark valleys) compared to maxima (bright peaks). Figure 4 (Full table in appendix 12) illustrates this phenomenon for our four representative models, revealing distinct tiers of reasoning ability and exposing different failure modes:

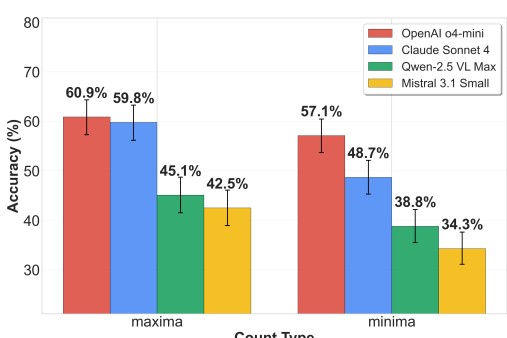

Figure 4: Analysis of accuracy by count type for the same models. This illustrates the performance gap when counting visually salient maxima versus less-salient minima. Error bars denote 95% confidence intervals.

**Robust Models:** The most capable models, **o4-mini** and **o3**, show only a minimal performance drop (under 4 and 3 points, respectively). This suggests they have a more abstract reasoning ability that is less reliant on the brightness of a feature and can identify peaks and valleys almost equally well.

**Sensitive Models:** A second tier is highly sensitive to feature type, with **Claude Sonnet-4**'s accuracy dropping 11 points and **Pixtral-Large**'s over 9. Their failure modes show a clear divide: for minima, they consistently undercount (errors of -1, -2, -3), suggesting they fail to perceive less-salient dark features. For maxima, however, their errors are more varied and include frequent overcounting (+1 errors for **Claude Sonnet-4** and **Mistral-Medium**), indicating a different failure mode related to confusion with highly salient features.

### 4.2 TRANSFORMATION RECOGNITION

The results for the Transformation Recognition task are summarized in Table 2. The overall accuracy scores reveal a clear performance hierarchy, with o4-mini leading significantly. However, a deeper diagnosis requires analyzing how model performance changes with task difficulty, for which the breakdown by visual style consistency serves as an excellent proxy. This analysis reveals three distinct and revealing failure profiles among the models.

**Robust Reasoners:** The top-performing models, like o4-mini, exhibit the expected behavior. Their accuracy is high but drops gracefully on the harder version of the task (e.g., by 13 points when plot types differ). This sensitivity to difficulty is the signature of a model genuinely engaging with the visual problem.

**Brittle Reasoners:** A second group of models, including Claude Sonnet 4 and Pixtral Large, are also sensitive to task difficulty, but their reasoning is fragile. Claude Sonnet 4's accuracy plummets by a catastrophic 19.5 points on the harder "Different Plot Type" task, with its performance falling well below that of a random guesser. This suggests a reasoning mechanism that is present but breaks completely under increased abstraction, a fundamentally different failure mode.

**Disengaged Heuristic Models:** The final group, most notably the LLaVA family, exhibits an anomalous insensitivity to task difficulty. Their performance remains perfectly flat, hovering just a few points above the 20% random-chance baseline. This is not robustness; it is a symptom of disengagement. A random guessing strategy is naturally insensitive to the difficulty of the questions. The LLaVA models' behavior is a strong indicator of a non-reasoning heuristic, where the model has defaulted to a strategy that is disconnected from the visual evidence.

These distinct patterns demonstrate the diagnostic power of the benchmark, revealing not just that models fail, but the specific ways in which their underlying reasoning strategies differ and break down.

Table 2: Performance on the Transformation Recognition task, with a breakdown by visual style consistency. The data reveals a clear performance drop for most models when plot styles differ, confirming an increase in task difficulty.

| Model | Overall Acc. (%) | Plot Type | | Color Map | |
|---|---|---|---|---|---|
| | | Same | Different | Same | Different |
| OpenAI o4-mini | $67.12 \pm 2.45$ | 72.31 | 59.2 | 72.4 | 66.16 |
| OpenAI o3 | $67.0 \pm 2.49$ | 71.55 | 57.96 | 69.32 | 67.93 |
| Claude Sonnet 4 | $29.0 \pm 2.48$ | 35.71 | 16.17 | 33.28 | 29.55 |
| Mistral 3.1 Small | $26.83 \pm 2.42$ | 27.57 | 25.37 | 29.38 | 23.99 |
| LLaVA 1.6 13B | $24.83 \pm 1.27$ | 24.81 | 24.88 | 25.49 | 26.26 |
| LLaVA 1.6 34B | $24.67 \pm 0.25$ | 24.56 | 24.88 | 23.38 | 25.25 |
| Pixtral Large | $16.58 \pm 2.42$ | 19.17 | 11.44 | 18.34 | 16.16 |
| Mistral 3.1 Medium | $14.5 \pm 2.37$ | 18.3 | 6.97 | 16.88 | 13.13 |
| Qwen-2.5 VL Max | $10.0 \pm 2.45$ | 11.9 | 6.22 | 10.88 | 10.86 |
| LLaVA 7B | $3.67 \pm 0.42$ | 4.26 | 2.49 | 3.9 | 2.78 |
| Random Baseline | 20 | 20 | 20 | 20 | 20 |

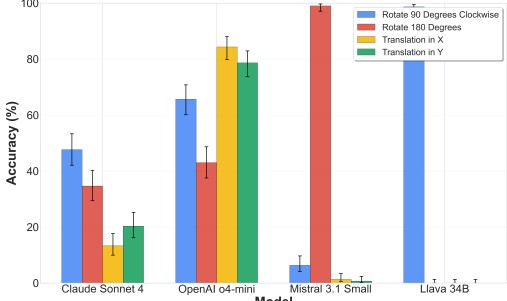

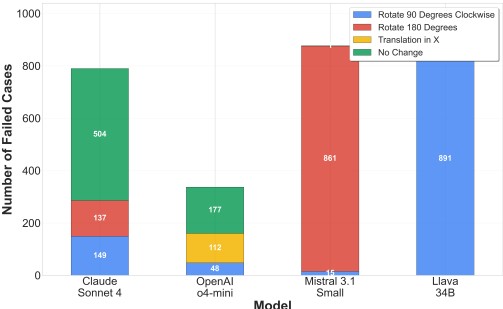

Figure 5: Strict accuracy by transformation option for four representative models. The plot reveals strong, systematic biases; some models fail at translation while excelling at a single rotation type.

Figure 6: Distribution of the top three incorrect answers for each model. The plot diagnoses distinct failure modes, from conservative guessing ("No Change") to collapsing to a single, predetermined answer.

### 4.3 ANALYSIS OF TRANSFORMATION-SPECIFIC BIASES AND FAILURE MODES.

A fine-grained analysis of performance by transformation option, combined with an analysis of the models' failure modes, reveals not just that models fail, but that they fail in profoundly different ways, exposing the chasm between heuristic-based and reasoning-based strategies.

The most dramatic finding is a pattern of degenerate, non-reasoning behavior in several models. As shown in Figure 5 (left) (full table in appendix 13), models like Mistral Small 3.1 and LLaVA-34b exhibit a paradoxical, "one-trick pony" performance. They are completely incapable of translation (scoring near 0%) but are nearly perfect at identifying a single, specific rotation type (99.0% for 180° rotations for Mistral, 98.67% for 90° rotations for LLaVA).

The reason for this behavior is not specialized competence, but a collapsed strategy. Figure 6 (right) provides the "smoking gun": when these models are wrong, their most frequent incorrect answer is their single preferred rotation type. This confirms their high accuracy is a direct artifact of them consistently guessing a predetermined answer, regardless of the visual input.

In stark contrast, the failure analysis reveals a fundamentally different and more sophisticated failure mode for the top-performing models. As seen in Figure 6, o4-mini the most common failure mode is the "No Change" option. This is the signature of a system that is genuinely engaging with the task: it attempts to find a transformation, fails to do so with high confidence, and defaults to the "null hypothesis." This is a failure within the process of reasoning, a sign of uncertainty rather than a predetermined guess, and it represents a qualitatively more advanced kind of error than the heuristic-based strategies employed by other models.

### 4.4 Correlation with Established Benchmarks

To demonstrate that the foundational skills isolated by MaRVL-QA are critical for real-world applications, we analyzed its relationship with established benchmarks that serve as proxies for complex, real-world reasoning. Our **Topological Counting** task showed strong, statistically significant correlations with both MMMU ($\rho = 0.85, p < 0.01$; N=9) and MathVista ($\rho = 0.85, p < 0.01$; N=8). Since these benchmarks test a wide spectrum of abilities from reasoning over statistical plots and geometric figures in MathVista to general scientific and expert-level understanding in MMMU, this strong correlation indicates that systematic enumeration is a foundational skill required for these complex applications. The **Transformation Recognition** task correlated powerfully with CharXiv ($\rho = 0.9, p < 0.05$; N=5), confirming that geometric reasoning is fundamental to chart comprehension. These findings indicate that MaRVL-QA performance is not only diagnostic but also predictive of downstream benchmark success.

## 5 Architectural Bottlenecks in Visual-Mathematical Reasoning

The systemic failures on MaRVL-QA are not incidental but symptomatic of core architectural bottlenecks in modern MLLMs, which are fundamentally optimized for semantic understanding at the expense of precise structural analysis. A primary bottleneck is the vision encoder's training objective; models like CLIP are optimized to align an entire image with a text caption, an objective that prioritizes capturing the semantic "gist" over preserving the fine-grained spatial detail required for procedural tasks like counting. This semantic compression is likely exacerbated by the lossy projection layers connecting the vision and language components, which discard the high-fidelity geometric information needed to distinguish between transformations. The language model, therefore, does not reason over the visual world itself but over a low-dimensional, pre-digested abstraction. This forces it to learn brittle heuristics - associating superficial changes in token patterns with transformation labels - rather than acquiring a true, generative understanding of geometric operations, leading to the collapsed strategies we observe.

## 6 Conclusion and Future Work

We introduced MaRVL-QA, a diagnostic benchmark designed to evaluate the foundational visual-mathematical skills of Multimodal Large Language Models. By testing models on topological counting and transformation recognition, tasks which isolate geometric reasoning from the semantic complexity of natural images, our work reveals profound fragility and a reliance on superficial heuristics in even state-of-the-art systems. The strong correlation between performance on MaRVL-QA and established multimodal benchmarks confirms that these foundational skills are critical for complex, real-world applications. The specific failure modes diagnosed by our benchmark illuminate a clear path forward, highlighting the need for architectures that enforce sequential processing to overcome counting failures, pre-training objectives grounded in geometric principles to teach true spatial reasoning, and aggressive data augmentation to ensure models learn to separate underlying structure from superficial style.

## 7 REPRODUCIBILITY STATEMENT

We are committed to full reproducibility and will publicly release all code and data used for benchmark generation and evaluation. The supplementary material includes a small data sample for both the tasks. The MaRVL-QA benchmark is programmatically generated from a curated library of 32 mathematical function families (Appendix A.1), ensuring the entire process is replicable. The generation pipeline (Section 3), including plot rendering (Appendix A.2) and task-specific ground truth creation with rigorous ambiguity filtering (Algorithm 1, Appendices A.3-A.5), is thoroughly documented. Our evaluation methodology (Section 4) is designed to be deterministic, with model temperatures set to 0 and the exact system prompts provided in Appendix A.7. To ensure fair and transparent answer parsing, we employ an LLM-based parser, with results from a stricter rule-based parser also reported for comparison (Appendix A.6.1). The public release will include all code necessary to generate the dataset, run model evaluations, parse results, and produce the figures and tables in this paper, facilitating the complete verification of our findings.

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

## A    APPENDIX

### A.1    FUNCTION FAMILY TIERS

The library is organized into five distinct tiers of complexity:

- **Tier 1: Foundational & Quadric Surfaces** This tier establishes a baseline with the most fundamental geometric forms. Key examples include the *Plane* and the *Elliptic Paraboloid*.
- **Tier 2: Periodic & Wave-like Surfaces** This tier introduces functions with regular, repeat-ing structures to test reasoning over periodic patterns. Examples include the *Wave Surface* and the *Ripple* function.
- **Tier 3: Singularities, Boundaries, & Discontinuities** This tier is designed to probe model robustness by presenting challenging edge cases. It contains functions like the *Cone* and the *Step Function* .
- **Tier 4: Composite & Modulated Surfaces** This tier features more complex topologies created by combining or modifying simpler forms. One example is the *Gaussian Mixture*, which allows for the precise placement of multiple, varied local extrema.
- **Tier 5: Advanced & Special Surfaces** The final tier includes advanced functions that represent highly complex geometric forms. Examples include the *Hyperboloid of One Sheet* and the *Extruded Witch of Agnesi*.

This tiered curation is a cornerstone of our methodology. It ensures that MaRVL-QA provides a rich, structured, and challenging environment for evaluating the visual reasoning capabilities of MLLMs.

Table 3 shows the number of functions used from each of the function families.

### A.2    PLOT RENDERING DETAILS

This section provides the specific implementation details for the plot rendering.

Table 3: Distribution of function instances across 32 families.

| Function Count | Family |
| --- | --- |
| 20 | Lattice of Gaussian Peaks, Inverted Lattice of Valleys |
| 17 | Windowed Waves |
| 13 | Gaussian Mixture, Inverted Gaussian Mixture |
| 12 | Crossed Tunnels |
| 10 | Plane, Elliptic Paraboloid, Hyperbolic Paraboloid, Parabolic Cylinder Wave Surface, Circular Ripples, Gabor Function, Checkerboard, Cone, Cusp, Monkey Saddle, Step Function, Sharp Ridge, Astroidal Bowl, Logarithmic Singularity, Ring Singularity, Modulated Wave, Damped Oscillator, Volcano, Spiraling Surface, Mixed-Sign Gaussian Mixture, Contained Symmetric Lobes,Beaded Rings, Inverted Beaded Rings, Hyperboloid of One Sheet, Extruded Witch of Agnesi |

**Plot Styles and Color Maps**   To ensure visual diversity and test for model robustness, each function was rendered using a combination of different plot types and color maps.

- **Plot Types:** Three distinct plot visualization styles were used:

    1. `heatmap`: A standard two-dimensional heatmap.
    2. `contour`: A plot showing only the contour lines.
    3. `heatmap-contour`: A plot overlaying the contour lines on top of the heatmap.

- **Color Maps:** All plot types were rendered using four different perceptually uniform and colorblind-friendly Matplotlib color maps:

    1. `viridis`
    2. `plasma`
    3. `inferno`
    4. `magma`

**Rendering and Analysis Resolution**   A distinction was made between the grid resolution used for rendering the final image and the one used for the numerical analysis to find ground-truth extrema.

- **Rendering Grid:** For generating the final plot images presented to the models, all functions were sampled over a 400x400 grid. This resolution was chosen to produce visually clear, high-quality images without being computationally prohibitive.

- **Analysis Grid:** For the initial numerical analysis in the Topological Counting task, a much denser 2000x2000 grid was used. This higher resolution was essential for the coarse localization stage, ensuring that the peak-finding algorithm could reliably detect even small or closely-spaced features before they were refined using numerical optimization on the underlying continuous function. As mentioned in the main paper, any potential disparities between the high-resolution analysis and the final rendered image were resolved during our manual curation phase, where we certified that the programmatic ground-truth count exactly matched the number of clearly visible features in the image.

## A.3   IMPLEMENTATION DETAILS FOR THE TOPOLOGICAL COUNTING TASK

This section provides the specific implementation details for the ground-truth generation pipeline of the Topological Counting task. The process involves a two-stage automated analysis followed by a manual curation protocol.

**Automated Numerical Analysis.**   The hybrid coarse-to-fine strategy for identifying extrema was implemented as follows:

1. **Coarse Localization:** An initial, efficient localization of extrema was performed on a discrete data grid using the `skimage.feature.peak_local_max` function to identify approximate pixel coordinates.

2. **High-Precision Refinement:** The pixel coordinates from the coarse localization step were used as initial seeds for a high-precision numerical optimization using the `scipy.optimize.minimize` function to refine the location of each extremum on the underlying continuous mathematical function.

Table 4 summarizes the specific parameters and functions used in this automated analysis.

Table 4: Parameters for Automated Extrema Detection.

| Stage | Parameter | Specification |
|---|---|---|
| Coarse Localization | Analysis Grid | $2000 \times 2000$ |
| | Peak Detection Function | `peak_local_max` |
| | `min_distance` | 10 pixels |
| | `exclude_border` | `True` |
| High-Precision Refinement | Optimization Function | `minimize` |
| | Method | 'Nelder-Mead' |
| | Tolerance (`tol`) | `1e-9` |
| | Final Precision | Rounded to 4 decimal places |

| Family Name | Countable Features |
|---|---|
| Wave Surface | Maxima & Minima |
| Lattice of Gaussian Peaks | Maxima only |
| Inverted Lattice of Valleys | Minima only |
| Gabor Function | Maxima & Minima |
| Crossed Tunnels | Minima only |
| Gaussian Mixture | Maxima only |
| Inverted Gaussian Mixture | Minima only |
| Mixed-Sign Gaussian Mixture | Maxima & Minima |
| Windowed Waves | Maxima & Minima |
| Contained Symmetric Lobes | Maxima & Minima |
| Beaded Rings | Maxima only |
| Inverted Beaded Rings | Minima only |

Table 5: Function families selected for the Topological Counting task and the specific features (maxima, minima, or both) certified for unambiguous counting.

**Manual Curation Protocol.** Following the automated analysis, a meticulous manual review was conducted by members of the research team to certify that every generated question was visually unambiguous and fair. The protocol for this review was as follows:

1. **Feature Visibility and Disambiguation:** Each identified extremum had to be a clearly perceptible, discrete feature. The criteria for this varied by plot type:
   - In heatmap plots, a maximum (or minimum) had to be visible as a distinct region of peak color intensity that faded before brightening again towards a neighboring maximum.
   - In contour plots, an extremum had to be enclosed by at least one contour line of lower (or higher) intensity relative to its immediate surroundings, ensuring it was not part of a flat plateau or ridge.

2. **Final Certification:** The reviewers confirmed that for every plot, the number of programmatically identified extrema exactly matched the number of features that met the visual criteria above. This final step ensured the integrity of the ground truth and its alignment with the visual evidence presented to the models.

---

**Algorithm 1** Unambiguous Transformation Generation

---

**Input**: A function $f$, a candidate transformation $T$.
**Output**: An unambiguous QA pair, or failure.

1: **if** $T$ is a rotation **then**
2:    **if** IsSymmetric($f, T$) **then**
3:       **return** failure {Reject symmetric rotation}
4:    **end if**
5:    $T_{equiv} \leftarrow$ FindEquivalentTranslation($f, T$)
6:    **if** IsSimilar($f, T, T_{equiv}$) **then**
7:       **return** failure {Reject if rotation resembles translation}
8:    **end if**
9: **else if** $T$ is a translation **then**
10:    **if** not HasProminentFeatures($f$) **then**
11:       **return** failure {Reject featureless surfaces}
12:    **end if**
13:    *found_pure_translation* $\leftarrow$ false
14:    **for** each candidate distance $d$ **do**
15:       Let $T_d$ be a translation by $d$.
16:       **if** IsSimilar($f, f, T_d$) **then**
17:          **continue** {Skip if translation is symmetric}
18:       **end if**
19:       **if** IsSimilarToAnyRotation($f, T_d$) **then**
20:          **continue** {Skip if translation resembles a rotation}
21:       **end if**
22:       $T \leftarrow T_d$
23:       *found_pure_translation* $\leftarrow$ true
24:       **break**
25:    **end for**
26:    **if** not *found_pure_translation* **then**
27:       **return** failure
28:    **end if**
29: **end if**
30: Generate QA pair for $(f, T)$.
31: **return** QA pair

---

### A.4 IMPLEMENTATION DETAILS FOR THE TRANSFORMATION RECOGNITION TASK

This section provides the specific implementation details for the ground-truth generation pipeline of the Transformation Recognition task.

**Ambiguity Filtering Algorithm.** At the core of our pipeline is a rigorous, two-way ambiguity filter designed to ensure that each transformation is visually distinct and not confusable with another. The high-level logic is detailed in Algorithm 1. The core of this algorithm is a similarity check, which we quantify by computing the normalized Root Mean Squared Error (NRMSE) on the functions' $2000 \times 2000$ sampled data grids. The specific hyperparameters used in this algorithm are detailed in Table 6.

Table 6: Hyperparameters for Ambiguity Filtering (Algorithm 1).

| Parameter | Value | Description |
|---|---|---|
| Comparison Grid | $2000 \times 2000$ | Resolution of data grids for NRMSE. |
| Significance Threshold | 0.07 | Min. NRMSE for a transformation to be non-symmetric. |
| Candidate Samples | 10 | Number of random translation distances tested per axis. |

**Prominent Feature Filtering for Translations.** To ensure that translation tasks were visually meaningful, we filtered out functions that lacked prominent features. This was a two-stage process:

1. **Automated Prominence Check:** We programmatically identified peaks and calculated their "prominence" (the vertical distance from the peak to the lowest point on any path to a higher peak). A function was flagged as having prominent features if at least one peak had a prominence of at least 10% of the function's total dynamic range.

2. **Manual Review:** Functions that failed the automated check were manually reviewed. This was crucial for functions with visually prominent features that are not peaks (e.g., sharp discontinuities or ridges). If such features were deemed significant by a human reviewer, the function was added back to the candidate pool for translation tasks.

**Combinatorial Style Expansion.** To test for robust, style-agnostic reasoning, we generated 81 visually distinct QA pairs for each validated (function, transformation) pair. This was achieved by creating a palette of 9 unique plot styles and then generating all possible pairings for the original and transformed plots. The 9 styles were derived from our plot types and color maps as follows:

- **Heatmap:** 4 styles (one for each color map: viridis, plasma, inferno, magma).
- **Contour with Heatmap:** 4 styles (one for each color map).
- **Contour-Only:** 1 style (contours are black and white, no color map applied).

This resulted in 9 styles (Plot A) $\times$ 9 styles (Plot B) $= 81$ unique visual combinations for every question.

## A.5 AMBIGUOUS CASES

Table 7: Examples of ambiguous transformation cases identified by our rigorous ambiguity detection algorithm. Each row displays an image with its corresponding question, actual answer, and the possible options that create the ambiguity.

| Image | Question | Actual Answer | Possible Options |
|---|---|---|---|
| **Section A: Symmetric Rotations** | | | |
|  | *Which single transformation has been applied to plot A to produce plot B?* | **Rotation 90 degrees clockwise** | Rotation 180 degrees No Change |
|  | *Which single transformation has been applied to plot A to produce plot B?* | **Rotation 180 degrees** | No Change |
| **Section B: Symmetric Translations** | | | |
|  | *Which single transformation has been applied to plot A to produce plot B?* | **Translation in X** | No Change |
|  | *Which single transformation has been applied to plot A to produce plot B?* | **Translation in Y** | No Change |
| **Section C: Rotations Appearing as Translations** | | | |
|  | *Which single transformation has been applied to plot A to produce plot B?* | **Rotation 180 degrees** | Translation in X |
|  | *Which single transformation has been applied to plot A to produce plot B?* | **Rotation 90 degrees clockwise** | Translation in Y |
| **Section D: Undetectable Translations for Featureless Functions** | | | |
|  | *Which single transformation has been applied to plot A to produce plot B?* | **Translation in Y** | No Change |
|  | *Which single transformation has been applied to plot A to produce plot B?* | **Translation in X** | No Change |

## A.6 Additional Analysis Data

### A.6.1 Results without LLM Parser

The tables below show the model performance when using a strict, rule-based parser instead of an LLM to extract answers. The results for most proprietary models that consistently adhered to the specified output format, such as OpenAI o4-mini and Claude Sonnet 4, show minimal to no change across both tasks.

However, the performance of several open-source models drops, revealing their tendency to fail to follow formatting instructions. The most dramatic impact is on the LLaVA models in the **Topological Counting task** (Table 8), where accuracies fall significantly. For example, LLaVA 13B's accuracy drops from 6.65% to 1.29%, indicating that its outputs were frequently valid in natural language (e.g., "twelve") but not in the required integer format ("12").

In contrast, the results for the **Transformation Recognition task** (Table 9) are much more stable. Most models successfully followed the multiple-choice format by outputting a single integer, resulting in negligible accuracy changes. This highlights that the models' formatting failures are task-dependent; they struggle more with open-ended numerical answers than with selecting from a constrained set of options.

Table 8: Overall performance on the Topological Counting task with rule-based extraction

| Model | Accuracy (%) |
|---|---|
| OpenAI o4-mini | 58.91 |
| Claude Sonnet 4 | 53.94 |
| OpenAI o3 | 50.06 |
| Qwen-2.5 VL Max | 41.80 |
| Pixtral Large | 37.6 |
| Mistral 3.1 Small | 38.18 |
| Mistral 3.1 Medium | 34.88 |
| Llava 1.6 13B | 1.29 |
| Llava 1.6 7B | 0.65 |
| Llava 1.6 34B | 1.81 |

Table 9: Overall performance on the Transformation Recognition task with rule-based extraction

| Model | Accuracy (%) |
|---|---|
| OpenAI o4-mini | 67.92 |
| OpenAI o3 | 63.83 |
| Claude Sonnet 4 | 29.17 |
| Mistral 3.1 Small | 26.83 |
| Llava 1.6 13B | 24.83 |
| Llava 1.6 34B | 24.67 |
| Pixtral Large | 16.58 |
| Mistral 3.1 Medium | 14.5 |
| Qwen-2.5 VL Max | 10.0 |
| Llava 1.6 7B | 3.67 |

### A.6.2 Analysis by Colormap

To evaluate model robustness against superficial visual features, we analyzed performance across four different colormaps. Figure 7 illustrates this for four representative models. The results show that top-performing models like o4-mini and Claude Sonnet-4 are largely robust to these stylistic variations, with their accuracy remaining stable within a narrow 4-point range regardless of the colormap used. This suggests they do not rely on simple color-based heuristics.

In contrast, some mid-tier models exhibit sensitivity to style. For instance, Mistral 3.1 Small's performance varies by over 7 percentage points depending on the colormap, dropping from 41.86% on 'magma' to 34.59% on 'plasma'. This performance variance suggests these models may be relying on more fragile, color-dependent heuristics rather than a style-agnostic method for identifying features. The full breakdown for all models is available in 11.

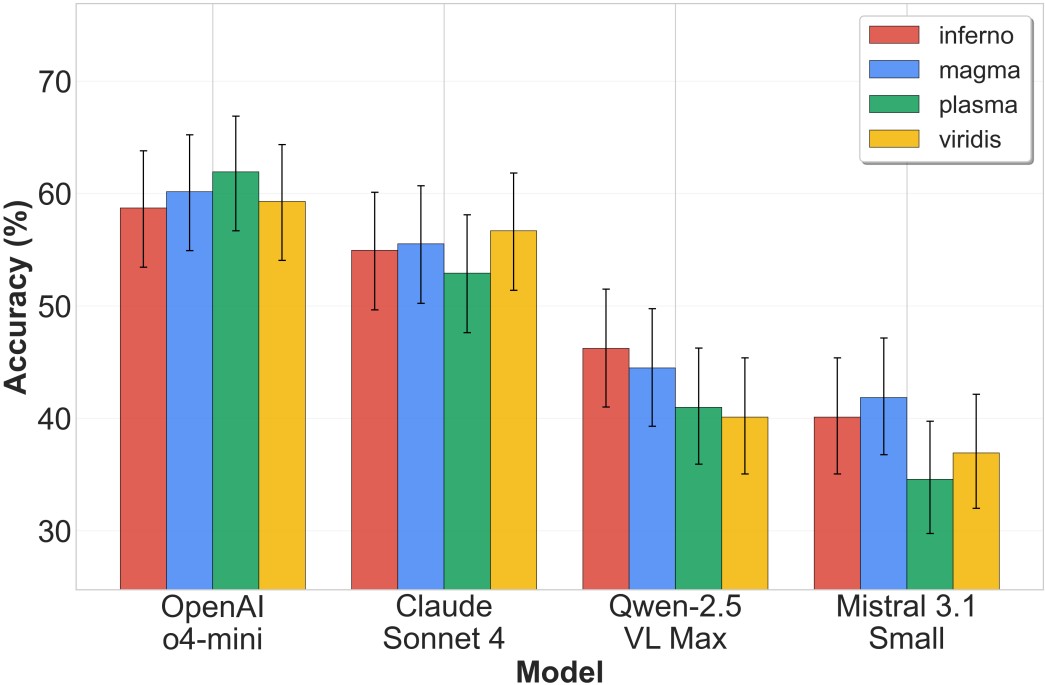

Figure 7: Analysis of accuracy by colormap for four representative models. The plot shows that top models are robust to stylistic variations. Error bars denote 95% confidence intervals.

Table 10: Detailed accuracy breakdown by feature count bin. We report Accuracy and Relaxed Accuracy at 10% and 20% tolerance. All metrics are shown with 95% confidence intervals.

| Model | Count Bin | Acc. (%) | Relaxed Acc. (10%) | Relaxed Acc. (20%) |
|---|---|---|---|---|
| OpenAI o4-mini | 1-5 | 69.64 | 69.64 | 74.80 |
| | 6-10 | 61.88 | 63.42 | 80.85 |
| | 11-15 | 32.22 | 53.70 | 71.11 |
| | 16-20 | 33.33 | 33.33 | 55.56 |
| Claude Sonnet 4 | 1-5 | 57.43 | 57.43 | 60.72 |
| | 6-10 | 60.85 | 62.39 | 76.58 |
| | 11-15 | 34.81 | 52.96 | 61.48 |
| | 16-20 | 33.33 | 33.33 | 44.44 |
| OpenAI o3 | 1-5 | 66.67 | 66.67 | 70.42 |
| | 6-10 | 45.98 | 48.72 | 62.56 |
| | 11-15 | 25.56 | 47.04 | 62.59 |
| | 16-20 | 33.33 | 33.33 | 57.41 |
| Qwen-2.5 VL Max | 1-5 | 50.86 | 50.86 | 57.12 |
| | 6-10 | 48.21 | 48.21 | 63.76 |
| | 11-15 | 10.00 | 17.41 | 45.56 |
| | 16-20 | 24.07 | 24.07 | 24.07 |
| Pixtral Large | 1-5 | 46.64 | 46.64 | 52.58 |
| | 6-10 | 40.34 | 41.37 | 64.10 |
| | 11-15 | 15.93 | 26.67 | 47.41 |
| | 16-20 | 31.48 | 31.48 | 35.19 |
| Mistral 3.1 Small | 1-5 | 44.44 | 44.44 | 52.43 |
| | 6-10 | 41.88 | 42.05 | 60.51 |
| | 11-15 | 17.04 | 31.11 | 49.63 |
| | 16-20 | 29.63 | 29.63 | 33.33 |
| Mistral 3.1 Medium | 1-5 | 48.83 | 48.83 | 57.28 |
| | 6-10 | 33.68 | 33.68 | 55.21 |
| | 11-15 | 10.37 | 17.78 | 39.63 |
| | 16-20 | 5.56 | 16.67 | 22.22 |
| Llava 13B | 1-5 | 15.18 | 15.18 | 19.09 |
| | 6-10 | 0.85 | 0.85 | 4.10 |
| | 11-15 | 0.37 | 0.37 | 2.22 |
| | 16-20 | 0.00 | 0.00 | 0.00 |
| Llava 7b | 1-5 | 1.56 | 1.56 | 1.56 |
| | 6-10 | 0.00 | 0.00 | 0.00 |
| | 11-15 | 0.00 | 0.00 | 0.00 |
| | 16-20 | 0.00 | 0.00 | 0.00 |
| Llava 34B | 1-5 | 0.31 | 0.31 | 0.31 |
| | 6-10 | 0.17 | 0.17 | 0.34 |
| | 11-15 | 0.00 | 0.00 | 0.00 |
| | 16-20 | 0.00 | 0.00 | 0.00 |

Table 11: Detailed strict accuracy (%) on the Topological Counting task by model and colormap.

| Model | Inferno (%) | Magma (%) | Plasma (%) | Viridis (%) |
|---|---|---|---|---|
| OpenAI o4-mini | 58.72 | 60.17 | 61.92 | 59.30 |
| Claude Sonnet 4 | 54.94 | 55.52 | 52.91 | 56.69 |
| OpenAI o3 | 52.91 | 49.13 | 47.97 | 54.36 |
| Qwen-2.5 VL Max | 46.22 | 44.48 | 40.99 | 40.12 |
| Pixtral Large | 44.19 | 39.83 | 38.66 | 35.76 |
| Mistral 3.1 Small | 40.12 | 41.86 | 34.59 | 36.92 |
| Mistral 3.1 Medium | 38.08 | 36.34 | 34.30 | 35.47 |
| Llava 13B | 5.23 | 6.69 | 5.81 | 8.43 |
| Llava 7b | 0.58 | 0.58 | 0.58 | 0.58 |
| Llava 34B | 0.00 | 0.00 | 0.29 | 0.29 |

Table 12: Detailed accuracy (%) on the Topological Counting task by feature type. All metrics are shown with 95% confidence intervals.

| Model | Maxima Acc. (%) | Minima Acc. (%) |
|---|---|---|
| OpenAI o4-mini | $60.91 \pm 3.47$ | $57.14 \pm 3.35$ |
| Claude Sonnet 4 | $59.81 \pm 3.50$ | $48.72 \pm 3.41$ |
| OpenAI o3 | $51.85 \pm 3.63$ | $49.33 \pm 3.42$ |
| Qwen-2.5 VL Max | $45.13 \pm 3.62$ | $38.83 \pm 3.34$ |
| Pixtral Large | $43.21 \pm 3.58$ | $34.07 \pm 3.28$ |
| Mistral 3.1 Small | $42.52 \pm 3.56$ | $34.31 \pm 3.29$ |
| Mistral 3.1 Medium | $40.05 \pm 3.51$ | $30.28 \pm 3.18$ |
| Llava 13B | $7.96 \pm 1.99$ | $5.49 \pm 1.59$ |
| Llava 7b | $1.37 \pm 0.88$ | $0.00 \pm 0.47$ |
| Llava 34B | $0.41 \pm 0.53$ | $0.00 \pm 0.47$ |

Table 13: Model Accuracy by Transformation Type with 95% Confidence Intervals.

| Model | Transformation Type | Accuracy (%) |
|---|---|---|
| Claude Sonnet 4 | Rotate 90 Degrees Clockwise | $47.7 \pm 5.6$ |
| | Rotate 180 Degrees | $34.7 \pm 5.4$ |
| | Translation in X | $13.3 \pm 3.8$ |
| | Translation in Y | $20.3 \pm 4.5$ |
| Llava 13B | Rotate 90 Degrees Clockwise | $14.0 \pm 3.9$ |
| | Rotate 180 Degrees | $0.0 \pm 0.6$ |
| | Translation in X | $85.3 \pm 4.0$ |
| | Translation in Y | $0.0 \pm 0.6$ |
| Llava 34B | Rotate 90 Degrees Clockwise | $98.7 \pm 1.4$ |
| | Rotate 180 Degrees | $0.0 \pm 0.6$ |
| | Translation in X | $0.0 \pm 0.6$ |
| | Translation in Y | $0.0 \pm 0.6$ |
| Llava 7b | Rotate 90 Degrees Clockwise | $13.3 \pm 3.9$ |
| | Rotate 180 Degrees | $0.0 \pm 0.6$ |
| | Translation in X | $1.3 \pm 1.4$ |
| | Translation in Y | $0.0 \pm 0.6$ |
| Mistral 3.1 Medium | Rotate 90 Degrees Clockwise | $19.0 \pm 4.4$ |
| | Rotate 180 Degrees | $34.7 \pm 5.4$ |
| | Translation in X | $1.7 \pm 1.6$ |
| | Translation in Y | $2.7 \pm 1.9$ |
| Mistral 3.1 Small | Rotate 90 Degrees Clockwise | $6.3 \pm 2.8$ |
| | Rotate 180 Degrees | $99.0 \pm 1.3$ |
| | Translation in X | $1.3 \pm 1.4$ |
| | Translation in Y | $0.7 \pm 1.1$ |
| OpenAI o3 | Rotate 90 Degrees Clockwise | $59.0 \pm 5.5$ |
| | Rotate 180 Degrees | $43.0 \pm 5.6$ |
| | Translation in X | $86.7 \pm 3.9$ |
| | Translation in Y | $79.3 \pm 4.6$ |
| OpenAI o4-mini | Rotate 90 Degrees Clockwise | $65.7 \pm 5.3$ |
| | Rotate 180 Degrees | $43.0 \pm 5.6$ |
| | Translation in X | $84.3 \pm 4.1$ |
| | Translation in Y | $78.7 \pm 4.6$ |
| Pixtral Large | Rotate 90 Degrees Clockwise | $3.3 \pm 2.1$ |
| | Rotate 180 Degrees | $48.0 \pm 5.6$ |
| | Translation in X | $7.0 \pm 2.9$ |
| | Translation in Y | $8.0 \pm 3.1$ |
| Qwen-2.5 VL Max | Rotate 90 Degrees Clockwise | $5.7 \pm 2.7$ |
| | Rotate 180 Degrees | $8.3 \pm 3.2$ |
| | Translation in X | $21.3 \pm 4.6$ |
| | Translation in Y | $4.7 \pm 2.4$ |

## A.7 System Prompts

We specify the exact system prompts used for every task to evaluate model responses.

---

**Counting System Prompt**

You are an expert in analyzing the topology of mathematical surfaces. Your primary function is to act as a feature counter for 2D plots of 3D functions, which will be presented as either heatmaps or contour plots. You will be asked to count the number of local maxima or local minima. When performing this task, you must adhere strictly to the following four rules for every plot you analyze:

1. Definition Rule: A feature must be a distinct peak (for a maximum) or valley (for a minimum).

2. Boundary Rule: Do not count a feature if its highest point (peak) or lowest point (valley) lies on the exact boundary of the plot area.

3. Plateau Rule: A single, continuous flat region (a plateau at a high value or a flat-bottomed basin at a low value) must be counted as exactly one feature.

4. Saddle Point Rule: You must not count saddle points. A saddle point is a location that appears to be a peak from some directions and a valley from others, and is not a true local extremum.

Your final answer for any task must be a single integer number (e.g., 0, 1, 2, etc.) in this format on a new line:

<final_answer>count</final_answer>

Here, `<final_answer>` and `</final_answer>` are XML tags and "count" is the integer number you counted. Do not provide any additional text, explanation, or justification.

---

**Transformation System Prompt**

You are an expert at comparing mathematical plots. You will be given a single input image containing two 2D plots of 3D functions - Plot A on the left and Plot B on the right.

Each plot can either be a heatmap or a contour plot. Your task is to determine the transformation that has been applied to Plot A (the left plot) to obtain Plot B (the right plot).

For transformations that include translation, the shift in any direction will not exceed 25% of the corresponding axis range (i.e. at most one-quarter of the plot's width or height).

You will also be given a set of numbered transformation options (labeled 1 ... N). Exactly one option is correct. Identify which single option describes the transformation applied to Plot A to obtain Plot B. If uncertain, pick the single closest option.

Output only one line in this exact format:

<final_answer>option_number</final_answer>

Here, `<final_answer>` and `</final_answer>` are XML tags and "option_number" is the number of the option you think is correct. Do not provide any additional text, explanation, or justification.

---

## A.8 Use of Generative AI

We utilized Generative AI tools to help improve the language, phrasing, and readability of this manuscript.

