# OpenReview forum: "MaRVL-QA: A Benchmark for Mathematical Reasoning over Visual Landscapes"
_ICLR.cc/2026/Conference — ICLR 2026 Conference Desk Rejected Submission_

### Official Review · Reviewer_kggk · 2025-10-26

**Soundness:** 2
**Presentation:** 3
**Contribution:** 2
**Rating:** 2
**Confidence:** 4

**Summary:**

The paper introduces MaRVL-QA, a diagnostic benchmark for visual–mathematical reasoning over function plots with two tasks: Topological Counting (count local extrema) and Transformation Recognition (identify a single applied rotation/translation). The authors generate >80k QA pairs with multi-stage ambiguity filtering, and release a balanced MaRVL-QA-Mini (1,548 counting items + 1,200 transformation items). Ten MLLMs are evaluated.

**Strengths:**

- MaRVL-QA provides a well-defined, controllable testbed for visual–mathematical reasoning over function plots.

- Two-way ambiguity filtering (e.g., excluding symmetric cases; distinguishing rotations vs. translations) and manual review improve label reliability.

- Confidence intervals are provided; format robustness is probed by comparing an LLM parser to a rule-based extractor.

**Weaknesses:**

- Synthetic, stylized plots may diverge from real scientific/engineering figures (e.g., noisy measurement fields, varied projections, non-uniform grids). The generated plots are mostly all in one distribution, which limited the representativeness of  MaRVL-QA.

- The benchmark is limited to a narrow set of skills, which might over-fit models toward a few primitives.

- The correlation analysis in Section 4.4 needs more explanation.

- Some important and recent models are missing from the baseline models, such as Gemini-2.5-pro.

- Some ablations are missing. For example, does the model performance improve if you add grids to the plot?

**Questions:**

- What is the performance of Gemini models on MaRVL-QA?

- How the correlation with other benchmarks are performed?

- Does fine-tuning helps model on such tasks?

---

> ### Author Response · Authors · 2025-11-26
> **Response to reviewer kggk - Part 1/3**
>
> We thank the reviewer for recognizing MaRVL-QA as a rigorous and controllable testbed, and for appreciating our robust ambiguity filtering and statistical validation methods.
>
> ### 1) Synthetic vs. Real Data
>
> > **Reviewer Comment:**
> > Synthetic, stylized plots may diverge from real scientific/engineering figures (e.g., noisy measurement fields, varied projections, non-uniform grids). The generated plots are mostly all in one distribution, which limited the representativeness of MaRVL-QA.
>
> We thank the reviewer for raising this crucial point about the difference between our stylized, synthetic plots and complex, real-world scientific and engineering figures (e.g., noisy measurement fields, varied projections). We acknowledge that MaRVL-QA currently samples from an idealized distribution of clean surface plots.
>
> **The Rationale: Diagnostic Isolation**
> This choice is a deliberate methodological necessity for a diagnostic benchmark. Our primary goal is diagnostic isolation, not immediate generalizability. Complex real-world data inherently contains two confounding sources of failure:
> * **Perceptual Noise:** Failures due to image noise, varied projections, and non-uniform grids.
> * **Foundational Reasoning Deficits:** Failures in the core tasks of structural enumeration and geometric equivariance.
>
> By using clean, semantically sparse visual landscapes, we effectively strip away the perceptual noise to isolate and precisely measure the foundational reasoning deficits that underpin all later failures. Our findings - where state-of-the-art models exhibit catastrophic failures on even these idealized, unambiguous tasks (e.g., "collapsed heuristics", enumeration breakdown) - confirm that these systems are not ready for the complexities of noisy real-world data.
>
> **Commitment to Future Generalization**
> We view performance on MaRVL-QA as a necessary prerequisite. Only once models reliably master the foundational, well-posed reasoning demonstrated by our clean plots can we effectively measure their robustness against the noise and complexities of noisy scientific figures from the real world. Future work will involve integrating noisy inputs and varied projections to fully test the generalizability of MLLMs.
>
> ---
>
> ### 2) Narrowness of the Benchmark
>
> > **Reviewer Comment:**
> > The benchmark is limited to a narrow set of skills, which might over-fit models toward a few primitives.
>
> We acknowledge the reviewer's concern that focusing on a narrow set of atomic primitives - Topological Counting and Transformation Recognition - might limit the general utility of the benchmark.
>
> **Narrowness of the benchmark is by design**
> The focused scope is a deliberate methodological strength designed for diagnostic isolation. We strip away semantic context to test the structural reasoning mechanisms directly. If models were overfitting to these primitives, we would expect near-perfect accuracy. Instead, our findings demonstrate profound fragility across all models. The top-performing model achieved only $58.91\%$ accuracy on the counting task, and other models relied on degenerate, non-reasoning heuristics (e.g., guessing a single rotation type). This outcome confirms that MLLMs are currently failing to generalize even within the benchmark's constrained structure, demonstrating a lack of robust foundational mechanism.
>
> **Proof of Transferability**
> Crucially, the skills isolated by MaRVL-QA are demonstrated to be foundational and transferable. We showed a strong, statistically significant Spearman Rank correlation between MaRVL-QA performance and scores on external, complex, high-level reasoning benchmarks: $\rho=0.85$ for MMMU and $\rho=0.85$ for MathVista. This correlation strongly suggests that the ability to perform systematic enumeration and geometric equivariance is an essential structural precursor that predicts success on broader, real-world reasoning tasks.

---

> > ### Author Response · Authors · 2025-11-26
> > **Response to reviewer kggk - Part 2/3**
> >
> > ### 3) Correlation Analysis
> >
> > > **Reviewer Comment:**
> > > The correlation analysis in Section 4.4 needs more explanation.
> >
> > We thank the reviewer for requesting clarification on the statistical measure used in Section 4.4. We report the Spearman Rank-Order Correlation Coefficient ($\rho$) to demonstrate the relationship between MLLM performance on MaRVL-QA and established external benchmarks.
> >
> > The use of Spearman's $\rho$ is crucial here because it is a non-parametric measure that assesses the strength and direction of a monotonic relationship between the ranks of models across two different benchmarks.
> > * **Spearman's $\rho$ Value (Strength of Correlation):** The resulting values of $\rho=0.85$ (for MMMU and MathVista) and $\rho=0.9$ (for CharXiv) indicate a nearly perfect correlation. This means that models ranked highly on MaRVL-QA (our benchmark) were also consistently ranked highly on complex, real-world reasoning tasks. This confirms that the foundational skills isolated by MaRVL-QA are highly predictive of overall MLLM capability.
> > * **$p$-Value (Statistical Significance):** The reported $p$-value of $p < 0.01$ signifies that the observed correlation is statistically significant, meaning there is less than a $1\%$ probability that this strong relationship occurred by random chance.
> >
> > This analysis validates that the intrinsic visual-mathematical skills measured by MaRVL-QA are foundational, transferable, and critical for success in diverse multimodal tasks.
> >
> > ---
> >
> > ### 4) Missing Ablations
> >
> > > **Reviewer Comment:**
> > > Some ablations are missing. For example, does the model performance improve if you add grids to the plot?
> >
> > We thank the reviewer for the insightful suggestion to further investigate the models' failure modes. To address the question of whether visual alignment or text reading is the primary bottleneck/crutch, we conducted two new ablation studies on a representative set of models. We evaluated performance under three conditions:
> > * **Baseline:** The original plots from our benchmark. We take a 200 example subset.
> > * **Grid Line Ablation:** Plots re-rendered with grid lines enabled.
> > * **Numerical Label Ablation:** Plots re-rendered with axis tick numbers removed to test reliance on textual cues.
> >
> > The results, summarized in the table below, demonstrate that the task's difficulty is rooted in reasoning about the geometric surface itself rather than visual parsing issues.
> >
> > | Model | Baseline (%) | Grid Lines (%) | Delta | No Numbers on Axis (%) | Delta |
> > | :--- | :--- | :--- | :--- | :--- | :--- |
> > | OpenAI o4-mini | 65.0 | 64.5 | (-0.5) | 56.5 | (-8.5) |
> > | OpenAI o3 | 60.0 | 59.5 | (-0.5) | 54.0 | (-6.0) |
> > | Claude Sonnet 4 | 33.5 | 35.5 | (+2.0) | 33.0 | (-0.5) |
> > | Pixtral | 18.0 | 17.5 | (-0.5) | 17.0 | (-1.0) |
> > | Mistral Medium | 17.0 | 17.0 | (0.0) | 17.0 | (0.0) |
> >
> > **Key Takeaways:**
> > * **Grid Lines:** As shown in the table, adding grid lines provides no meaningful improvement across any of the tested models. The performance changes are negligible (e.g., -0.5% for o4-mini and +2.0% for Claude Sonnet 4), which confirms that the models' failures are not due to an inability to visually align features with the axes.
> > * **Numerical Labels:** Removing axis numbers caused a moderate performance drop for top-tier models (e.g., o4-mini dropped by 8.5%), while lower-performing models showed almost no change. This indicates that while state-of-the-art models may leverage numerical grounding to some extent, the task cannot be solved by OCR tricks, and the core challenge remains geometric reasoning.
> >
> > These results reinforce our claim that MaRVL-QA isolates a fundamental deficit in geometric and topological reasoning, a challenge that cannot be mitigated by adding simple visual aids.
> >
> > ---

---

> > > ### Author Response · Authors · 2025-11-26
> > > **Response to reviewer kggk - Part 3/3**
> > >
> > > ### 5) Fine-tuning
> > >
> > > > **Reviewer Comment:**
> > > > Does fine-tuning helps model on such tasks?
> > >
> > > We intentionally did not fine-tune the MLLMs, as our objective was to diagnose their zero-shot ability to ground abstract mathematical concepts. However, to rule out visual ambiguity as a confounding factor, we trained a supervised ResNet50 baseline, which confirms that the necessary discriminative features are present and learnable.
> > >
> > > We trained a Computer Vision baseline using ResNet50 (initialized using pre-trained weights) on the Transformation Recognition task. This established a performance ceiling for a model focused purely on visual classification, allowing us to isolate the MLLMs' zero-shot reasoning deficiencies.
> > >
> > > **Model and Training Details:**
> > > * **Model:** ResNet50 Backbone (~24 million parameters).
> > > * **Dataset Split (Zero Function Overlap):**
> > >     * **Training Set:** 58,725 examples from 279 unique functions.
> > >     * **Testing Set:** 388 examples from 38 separate, held-out unique functions. The test set was strategically sampled to ensure representation across all plot types and applicable transformations.
> > > * **Training Hyperparameters:** Trained for 50 epochs (with loss converging in ~10 epochs), using a batch size of 128, a learning rate of 1e-3 and Adam optimizer with ReduceLROnPlateau scheduler.
> > >
> > > **Comparative Results:**
> > >
> > > | Model Type | Model Name | Training Paradigm | Accuracy (%) |
> > > | :--- | :--- | :--- | :--- |
> > > | **Pure Vision** | **ResNet50** | **Supervised Training** | **87.11** |
> > > | Multimodal | O4-mini | Zero-Shot | 57.73 |
> > > | Multimodal | O3 | Zero-Shot | 49.74 |
> > > | Multimodal | Sonnet-4 | Zero-Shot | 24.74 |
> > >
> > > **Justification: Strengthening the MaRVL-QA Baseline**
> > > The large gap between the fine-tuned ResNet50 and the zero-shot MLLMs is highly informative and directly strengthens our paper's core argument:
> > >
> > > 1.  **Confirmed Task Solvability**
> > >     The high accuracy of the supervised ResNet50 (87.11%) establishes a visual feasibility oracle, proving the task is visually solvable and eliminating the possibility that MLLM failures are due to visually ambiguous data or inherent task difficulty.
> > > 2.  **Isolating the Reasoning Gap**
> > >     The MLLMs' zero-shot failure is, therefore, not a simple "training mismatch" for visual classification. If the MLLMs' vision modules were merely sub-optimal, their performance would still be expected to be significantly higher given the simplicity of the visual task and their vast pre-training. Instead, the persistent performance gap points to a failure to perform zero-shot generalization. The MLLMs fail to successfully:
> > >     * Generalize abstract mathematical concepts (their symbolic knowledge) to a novel visual domain (the plots).
> > >     * Couple their visual observation with the correct abstract textual label.
> > >
> > > This confirms that MaRVL-QA successfully isolates a genuine mathematical-visual reasoning bottleneck and is a valuable benchmark for testing abstract knowledge transfer in MLLMs.
> > >
> > > We will update the paper to include these results and the corresponding discussion.

---

### Official Review · Reviewer_azUV · 2025-10-27

**Soundness:** 4
**Presentation:** 3
**Contribution:** 3
**Rating:** 8
**Confidence:** 4

**Summary:**

The paper proposes MaRVL to quantitatively evaluate the ability of performing structured spatial analysis in MLLMs. The benchmark comprises two major types of tasks: (i) Topological counting, to test MLLMs on their ability to identify and enumerate local extrema and (ii) Transformation Recognition, to test their ability to detect applied geometric transformations. The authors also contribute a curated library of 80000 QA pairs of mathematical functions with multi-stage ambiguity filtering. Moreover, they analyse systematic failure modes in MLLMs such as the "one-trick pony" performance for smaller models such as LLaVA, while top-performing models such as o4-mini demonstrates a higher accuracy of 67.12% on the transformation recognition tasks.

**Strengths:**

The paper was clear and fluid to understand. Majorly, the strengths of the paper include:

1) Failure mode analysis: The study's dissection of failure modes - catastrophic, near-miss, heuristic collapse and bias profiles - maxima and minima salience, rotation, translation confusion.
2) Ambiguity Filtering: The paper employs algorithmic ambiguity filtering based on normalised RMSE thresholds, explicit rejection of symmetric or visually confounding transformations, and prominence-based feature validation.

**Weaknesses:**

Mainly, I have one weakness to point out, which is not necessary to fulfil in immediacy:

1) Synthetic Task Design: The benchmark relies on programmatically designed mathematical surface plots, which cover most aspects of visual reasoning, but lack in noise and the complexities of real-world visual reasoning.

**Questions:**

Clarification required in one aspect:

Limited Statistical and Error Robustness Analysis: While mean accuracies and confidence intervals are reported, the paper lacks statistical significance testing between models. It does not analyse potential dataset biases, such as the correlation between surface type. This leaves some uncertainty about whether observed differences are robust or dataset-specific.

---

> ### Author Response · Authors · 2025-11-26
> **Response to reviewer azUV**
>
> We thank the reviewer for appreciating the clarity of our presentation, the depth of our failure mode analysis, and the rigorous algorithmic ambiguity filtering employed to ensure benchmark integrity.
>
> > **Reviewer Comment:**
> > **Limited Statistical and Error Robustness Analysis:** While mean accuracies and confidence intervals are reported, the paper lacks statistical significance testing between models. It does not analyse potential dataset biases, such as the correlation between surface type. This leaves some uncertainty about whether observed differences are robust or dataset-specific.
>
> We appreciate the reviewer’s insistence on a rigorous statistical foundation and detailed robustness analysis. We have executed the requested analyses, and the results provide critical validation of the benchmark's diagnostic power and our analysis of model failure modes.
>
> ### 1. Statistical Significance of Model Differences
>
> We conducted McNemar’s tests on the binary outcomes for the full test sets, which reveal distinct performance relationships for each task.
>
> * **Topological Counting:** The performance difference between the top-ranked models is not statistically significant. The comparison of o4-mini ($58.91\%$) vs. Claude Sonnet 4 ($53.94\%$) yielded a p-value of $0.7681$. This result is central to our argument: the task is fundamentally unsolved, and the counting ability of these frontier models is statistically equivalent and equally limited, pointing to a uniform, deep-seated architectural flaw in procedural enumeration .
> * **Transformation Recognition:** Validation of Distinct Failure Modes. In contrast, this task reveals sharp, statistically distinct capability tiers.
>     * The difference between o4-mini ($67.12\%$) and Claude Sonnet 4 ($29.0\%$) is highly statistically significant (McNemar's $p$-value $= 0.0$). This validates the separation between Robust Reasoners and Brittle Reasoners, confirming o4-mini's strategy is genuinely more robust under visual abstraction.
>     * The difference between o4-mini ($67.12\%$) and o3 ($67.0\%$) is not statistically significant (McNemar's statistic $= 0.667$, p-value $= 0.4142$). This confirms equivalent geometric reasoning within models of the same family.
>
> ---
>
> ### 2. Analysis of Robustness to Surface Complexity (Dataset Bias)
>
> The reviewer questioned the correlation between performance and surface complexity. Our analysis across the five function family tiers (Appendix A.1) confirms that performance is not a dataset bias, but rather a fundamental lack of generalization when surface structure becomes irregular.
>
> **A. Topological Counting Collapse**
> The data below shows a severe, systemic collapse in accuracy when moving from structurally simple to complex surfaces.
>
> | Function Family Tier | O4-mini (Acc. %) | Sonnet 4 (Acc. %) | Qwen 2.5 VL Max (Acc. %) | Mistral 3.1 Small (Acc. %) |
> | :--- | :--- | :--- | :--- | :--- |
> | Periodic & Wave-like Surfaces | $86.54$ | $86.54$ | $78.85$ | $76.9$ |
> | Composite & Modulated Surfaces | $42.68$ | $36.59$ | $34.15$ | $17.07$ |
>
> On highly regular Periodic surfaces, models perform near-optimally ($>86\%$ for the top two). However, accuracy plummets by $43.86$ points for o4-mini on Composite & Modulated Surfaces (e.g., Gaussian Mixtures). This confirms models rely on learned heuristics optimized for regular, repetitive structures, rather than procedural counting that generalizes to complex or novel topologies.
>
> **B. Transformation Recognition and Feature Prominence**
> The complexity tiers similarly define distinct failure envelopes for the Transformation task.
>
> | Function Family Tier | O4-mini (Acc. %) | Sonnet 4 (Acc. %) | Mistral 3.1 Small (Acc. %) | LLaVA 34B (Acc. %) |
> | :--- | :--- | :--- | :--- | :--- |
> | Foundational & Quadric Surfaces | $71.05$ | $44.74$ | $26.32$ | $28.95$ |
> | Periodic & Wave-like Surfaces | $79.45$ | $32.88$ | $27.40$ | $24.66$ |
> | Singularities, Boundaries, & Discontinuities | $82.09$ | $22.39$ | $17.91$ | $23.88$ |
> | Composite & Modulated Surfaces | $71.54$ | $22.76$ | $13.01$ | $17.89$ |
> | Advanced & Special Surfaces | $57.89$ | $31.58$ | $31.58$ | $36.84$ |
>
> * **Robust Models:** o4-mini excels on surfaces with clear, distinct features (Singularities: $82.09\%$) but sees a significant drop on the most abstract surfaces (Advanced: $57.89\%$).
> * **Heuristic Models (LLaVA/Mistral Small):** These models, which rely on single-rotation heuristics (Section 4.3), show no consistent pattern relative to complexity, confirming their disengagement from the visual features of the surface.

---

### Official Review · Reviewer_gLLJ · 2025-10-31

**Soundness:** 3
**Presentation:** 3
**Contribution:** 3
**Rating:** 6
**Confidence:** 4

**Summary:**

This paper introduces MaRVL-QA (Mathematical Reasoning over Visual Landscapes), a novel diagnostic benchmark designed to evaluate the foundational visual-mathematical reasoning skills of Multimodal Large Language Models (MLLMs). The authors posit that current MLLMs excel at high-level semantic description but fail at a more fundamental level of precise spatial and structural analysis. To test this, the benchmark uses mathematical surface plots (e.g., 2D heatmaps of 3D functions) as a testbed, which isolates structural reasoning from the semantic "noise" of natural images.

MaRVL-QA consists of two core tasks:

1. Topological Counting: Requiring models to identify and enumerate local extrema (maxima or minima) from a single plot.
2. Transformation Recognition: Requiring models to identify which geometric transformation (e.g., translation, rotation) maps one plot to another.

The benchmark is programmatically generated from a curated library of mathematical functions, employing a rigorous, multi-stage ambiguity filtering process to ensure objective ground truth. Critically, tasks are designed to be "OCR-proof" (e.g., by holding axis labels constant) to force genuine geometric reasoning.

Evaluations on 10 SOTA MLLMs reveal profound deficits. The paper demonstrates that models (1) fail to scale as feature counts increase, (2) are biased by visual salience (e.g., better at counting bright "maxima" than dark "minima"), and (3) often resort to "collapsed" non-reasoning heuristics, such as always guessing the same rotation type regardless of the visual input. The paper concludes that these foundational skills are critical, showing strong correlations between MaRVL-QA performance and scores on complex benchmarks like MMMU and MathVista.

**Strengths:**

* The benchmark's premise is highly original and insightful. While many benchmarks test math reasoning (e.g., GSM8K) or chart QA (e.g., ChartQA), MaRVL-QA is the first I have seen to so effectively isolate foundational geometric and topological reasoning from high-level semantics. Using semantically-sparse "visual landscapes" as a diagnostic tool is a novel and powerful idea.

* The methodological quality of the benchmark's construction is a significant strength. The decision to hold axis labels constant in the Transformation Recognition task is a critical and intelligent design choice. It successfully preempts models from "cheating" by simply reading and comparing axis values, forcing them to reason about the surface geometry itself. The paper details a rigorous, multi-stage process to ensure objective ground truth. This includes (1) excluding ambiguous topological features like saddle points, (2) manually curating all counting questions to ensure features are visually unambiguous, and (3) programmatically filtering out confounding transformations (e.g., a rotation that looks like a translation, or a symmetric rotation that results in no change). This rigor is commendable. The benchmark is not static; it's generated from a curated library of 32 function families. Furthermore, it tests robustness by rendering plots across diverse styles (heatmaps, contours) and colormaps. The released MaRVL-QA-Mini test set is not just a random sample; it is a carefully stratified subset, balanced for factors like visual style consistency and transformation type, which enables fair and robust analysis.

* The paper is exceptionally well-written. The problem statement is clear, the methodology is justified at every step, and the results are presented effectively. Figures 1, 3, 5, and 6, in particular, do an excellent job of visualizing the tasks and the models' systematic, heuristic-based failures.

* The paper's findings are highly significant for the MLLM community.
    * MaRVL-QA serves as a powerful diagnostic tool. It moves beyond "models are bad at counting" to show *how* they fail. The discovery of "collapsed strategies" (e.g., Mistral Small's 99.0% accuracy on 180-degree rotations and ~0% on translations) is a "smoking gun" that proves a complete lack of generalizable reasoning.
    * The paper expertly identifies and distinguishes different failure modes: "Robust Reasoners" (O4-mini) that fail gracefully, "Brittle Reasoners" (Claude Sonnet 4) that collapse under difficulty, and "Disengaged Heuristic Models" (LLaVA) that are not reasoning at all.
    * The analysis in Section 5 provides a clear, plausible hypothesis for these failures, linking them to architectural bottlenecks where vision encoders trained for semantic "gist" (like CLIP) discard the fine-grained geometric detail needed for these tasks.
    * The strong, significant correlations with established benchmarks like MMMU (ρ=0.85) and MathVista (ρ=0.85) make a compelling case that these foundational skills are predictive of success on complex, real-world reasoning tasks.

**Weaknesses:**

* The evaluation includes a strong set of SOTA closed-source models but is very weak on the open-source side. It primarily features the LLaVA family (which are now several years old and known to be poor at this) and one Qwen model. The LLaVA models perform at or near random chance, offering little insight beyond "they can't do this at all." Including a wider range of modern, capable open-source MLLMs (e.g., InternVL, newer Llama-V) would be necessary to claim these failures are universal and not just an artifact of older architectures.

* The paper strongly justifies using an LLM (GPT-4.1) as a parser, stating rule-based parsers are "brittle" and models "failed to consistently adhere to the requested format". However, the appendix (A.6.1) shows that for the Transformation Recognition task, the results with a strict rule-based parser are "much more stable" with "negligible accuracy changes". The brittleness only seems to dramatically affect the LLaVA models on the counting task. This makes the justification in the main paper feel slightly overstated.

* The paper brilliantly analyzes the failure modes of low-performing models (e.g., LLaVA, Mistral Small). However, it only analyzes the *statistics* of the best model, O4-mini (e.g., its errors are "near-misses" and its most common error is "No Change"). It never explores *why* O4-mini fails. When it makes a "near-miss" count, is it consistently missing low-salience minima? When it guesses "No Change," is it on visually subtle transformations? This is a missed opportunity to understand the failure modes of competent models.

**Questions:**

1. The correlation analysis in Section 4.4 is a key part of your significance argument. However, the sample sizes (N) are very small: N=9 for MMMU, N=8 for MathVista, and N=5 for CharXiv. Given that 10 models were evaluated, could you clarify why the N-values are not 10? Is this due to public scores for some models being unavailable on those external benchmarks?

2. The analysis of "collapsed strategies" in weaker models is a highlight of the paper. Could you provide any qualitative insight into the failure modes of the strongest model, O4-mini? For instance, when it fails on Topological Counting with a "near-miss," does it exhibit a pattern, such as consistently missing features near the boundary or features with lower visual salience (minima)?

3. The open-source evaluation focuses heavily on the LLaVA family, which performs at chance level. This demonstrates a floor, but doesn't tell us much about capable open models. Have you considered evaluating more recent open-source MLLMs to see if they also adopt these heuristic strategies or if any are beginning to show "brittle" or "robust" reasoning profiles?

4. You provide a clear and strong justification for excluding saddle points due to their lack of a "clear visual signature". As a forward-looking question, do you believe this is a permanent limitation of 2D representations, or could a model trained specifically on the mathematical definition of a saddle point learn to identify them from contour plots, even when they are visually ambiguous to humans?

---

> ### Author Response · Authors · 2025-11-26
> **Response to reviewer gLLj - Part 1/3**
>
> We sincerely thank the reviewer for recognizing the novelty of our semantically-sparse diagnostic approach, the rigor of our ambiguity-filtered construction, and the significance of our findings in exposing fundamental architectural bottlenecks.
>
> ### 1) Open-Source Evaluation
>
> > **Reviewer Comment:**
> > The evaluation includes a strong set of SOTA closed-source models but is very weak on the open-source side. It primarily features the LLaVA family (which are now several years old and known to be poor at this) and one Qwen model. The LLaVA models perform at or near random chance, offering little insight beyond "they can't do this at all." Including a wider range of modern, capable open-source MLLMs (e.g., InternVL, newer Llama-V) would be necessary to claim these failures are universal and not just an artifact of older architectures.
> >
> > The open-source evaluation focuses heavily on the LLaVA family, which performs at chance level. This demonstrates a floor, but doesn't tell us much about capable open models. Have you considered evaluating more recent open-source MLLMs to see if they also adopt these heuristic strategies or if any are beginning to show "brittle" or "robust" reasoning profiles?
>
> We thank the reviewer for the insightful comment regarding the need to broaden our open-source model analysis to ensure our claims of systemic failure are universally applicable and not merely an artifact of dated architectures. We agree this is a critical point for demonstrating the generality of the MaRVL-QA diagnostic.
> To directly address this concern, we conducted new experiments evaluating six contemporary, state-of-the-art open-source Multimodal Large Language Models, including the powerful Qwen 3 (8B and 235B variants), the Gemma 3 family (12B, and 27B), and Microsoft Phi 4 Multimodal Instruct.
>
> The new results, summarized in the table below, strongly reinforce our central thesis that a fundamental, architectural deficit in systematic visual-mathematical reasoning persists across modern MLLMs.
>
> | Model | Topological Counting (Accuracy %) | Transformation Recognition (Accuracy %) |
> | :--- | :--- | :--- |
> | Qwen 3 235B | 44.64% | 25.83% |
> | Qwen 3 8B | 31.59% | 20.33% |
> | Gemma 3 27B | 16.93% | 19.92% |
> | Gemma 3 12B | 16.27% | 23.50% |
> | Phi 4 Multimodal Instruct | 18.86% | 23.67% |
> | Random Baseline | N/A | 20.00% |
>
> * **Topological Counting:** While Qwen 3 235B performs in a "brittle" tier ($44.64\%$), it still falls significantly short of the top proprietary models ($58.91\%$ for OpenAI o4-mini). The Gemma family and Phi Multimodal score in the $15-19\%$ range, confirming that catastrophic breakdown in counting remains a pervasive failure mode for the many of new open-source systems.
> * **Transformation Recognition:** Nearly all newly tested open-source models (including all Gemma variants and Qwen 3 8B/235B) perform at or near the $20\%$ random chance baseline. This is a decisive finding, indicating a complete absence of reliable, style-agnostic geometric reasoning, thus broadening our conclusion that the capability is fundamentally missing across architectures.
>
> ---
>
> ### 2) Parser Justification
>
> > **Reviewer Comment:**
> > The paper strongly justifies using an LLM (GPT-4.1) as a parser, stating rule-based parsers are "brittle" and models "failed to consistently adhere to the requested format". However, the appendix (A.6.1) shows that for the Transformation Recognition task, the results with a strict rule-based parser are "much more stable" with "negligible accuracy changes". The brittleness only seems to dramatically affect the LLaVA models on the counting task. This makes the justification in the main paper feel slightly overstated.
>
> We thank the reviewer for identifying the overstatement regarding the LLM parser's necessity for the Transformation Recognition task; we agree to revise the main text accordingly. The parser's primary methodological rationale was correcting answer format non-adherence in the Topological Counting task. Since counting requires an open-ended numerical response, lower-performing open-source models often output natural language (e.g., "twelve") instead of the required integer format.
>
> ---

---

> > ### Author Response · Authors · 2025-11-26
> > **Response to reviewer gLLj - Part 2/3**
> >
> > ### 3) Failure Mode Analysis of o4-mini
> >
> > > **Reviewer Comment:**
> > > The paper brilliantly analyzes the failure modes of low-performing models (e.g., LLaVA, Mistral Small). However, it only analyzes the statistics of the best model, O4-mini (e.g., its errors are "near-misses" and its most common error is "No Change"). It never explores why O4-mini fails. When it makes a "near-miss" count, is it consistently missing low-salience minima? When it guesses "No Change," is it on visually subtle transformations? This is a missed opportunity to understand the failure modes of competent models.
> > >
> > > The analysis of "collapsed strategies" in weaker models is a highlight of the paper. Could you provide any qualitative insight into the failure modes of the strongest model, O4-mini? For instance, when it fails on Topological Counting with a "near-miss", does it exhibit a pattern, such as consistently missing features near the boundary or features with lower visual salience (minima)?
> >
> > We thank the reviewer for requesting a deeper diagnostic analysis of o4-mini’s failures, which is essential for understanding the precise limits of competent models. We introduce a new analysis in the final paper focusing on the model’s procedural limits and graceful breakdown points.
> >
> > We clarify that the hypothesis that features near the plot boundary drive failure is accounted for in our rigorous data generation pipeline. As detailed in the appendix, our automated process programmatically excludes features near the border, and this was certified via manual curation to ensure every visible feature is unambiguously contained within the plotting area. Boundary proximity is therefore not a factor in o4-mini’s observed errors.
> >
> > The true limitations of o4-mini lie in two areas identified by our analysis of its errors:
> >
> > * **Procedural Limits (Visual Saliency Mitigation):** We performed an in-depth analysis of O4-mini's non-catastrophic 'near-miss' errors (those within 20% tolerance) on the Topological Counting task. The percentages we report relate the number of errors made on a feature type to the total 261 near-miss errors (o4-mini made 138 errors on minima samples and 123 on maxima samples). We find that o4-mini's subtle errors are distributed almost evenly (52.87% minima vs. 47.13% maxima). This demonstrates that o4-mini is not affected by visual saliency; its remaining errors are attributed to procedural miscounts rather than a structural perceptual flaw.
> > * **Graceful Breakdown (Geometric Orientation):** In Transformation Recognition, we analyze the cases where o4-mini failed and defaulted to its 'No Change' safety heuristic. We find that the true transformation for 68.4% of these failures was Rotation ($180^{\circ}$ and $90^{\circ}$ combined), while Translations accounted for 31.6%. This confirms that o4-mini's graceful failure is triggered not by small positional shifts (translation), but by the difficulty of reliably detecting a change in global orientation when the underlying surface structure remains unchanged within the visualized domain.
> >
> > We will incorporate analysis to substantiate these findings in the final version of the paper.
> >
> > ---
> >
> > ### 4) Correlation Analysis Sample Size
> >
> > > **Reviewer Comment:**
> > > The correlation analysis in Section 4.4 is a key part of your significance argument. However, the sample sizes (N) are very small: N=9 for MMMU, N=8 for MathVista, and N=5 for CharXiv. Given that 10 models were evaluated, could you clarify why the N-values are not 10? Is this due to public scores for some models being unavailable on those external benchmarks?
> >
> > We confirm that the varying sample sizes (N=9 for MMMU, N=8 for MathVista, N=5 for CharXiv) are solely due to the unavailability of publicly reported scores for some evaluated MLLMs on these external benchmarks, ensuring the analysis includes every model for which comparable data was sourced.
> >
> > ---

---

> > > ### Author Response · Authors · 2025-11-26
> > > **Response to reviewer gLLj - Part 3/3**
> > >
> > > ### 5) Saddle Points and Future Direction
> > >
> > > > **Reviewer Comment:**
> > > > You provide a clear and strong justification for excluding saddle points due to their lack of a "clear visual signature". As a forward-looking question, do you believe this is a permanent limitation of 2D representations, or could a model trained specifically on the mathematical definition of a saddle point learn to identify them from contour plots, even when they are visually ambiguous to humans?
> > >
> > > We think that high-resolution input offers a potential path to identifying saddle points, as it allows the MLLM to move from visual perception to simulated computation.
> > >
> > > This approach assumes the model can analyze minute changes in color and contour intensity to approximate the function's $2^{nd}$ order derivative information (the Hessian matrix), thereby grounding the detection in the formal mathematical definition rather than ambiguous visual signatures. This overcomes the key reason we excluded saddle points from our benchmark.
> > >
> > > However, this capability is currently limited by architectural bottlenecks: MLLMs are primarily optimized for semantic compression and suffer from lossy projection layers that discard the fine-grained geometric data necessary for calculus approximation. Therefore, while high-resolution input is necessary, the MLLM must first evolve architecturally to retain and reason over this precise, computational data. This simulated analysis defines the future frontier for robust visual-mathematical MLLMs.

---

### Official Review · Reviewer_FBeL · 2025-11-02

**Soundness:** 3
**Presentation:** 2
**Contribution:** 2
**Rating:** 2
**Confidence:** 5

**Summary:**

The paper introduces MaRVL-QA, a visual mathematical reasoning benchmark of function plots. It contains two tasks:

1. Topological Counting: count local maxima/minima from a plot
2. Transformation Recognition: identify the rotation/translation maps an original plot to a transformed one (5 multiple choice)

The authors describe a synthetic corpus (1,548 QA items for Topological Counting and 79,542 for Transformation Recognition) and a smaller MaRVL-QA-Mini subset (2,748 items). They evaluate ten MLLMs in zero shot mode with deterministic decoding (tempreture = 0), and argue that models rely on brittle heuristics rather than robust geometric reasoning. A full dataset and code release is promised but not yet included.

**Strengths:**

1. Great idea overall.
2. Clear task definitions.
3. Good analysis of model behavior: The paper doesn’t just report accuracy; it actually looks at how models fail. For example, some models always guess the same rotation or default to “No Change” when uncertain. Those observations clearly show where reasoning breaks down and make the results more insightful.
4. Potential for extensibility: The idea (function-plot generator) could produce many more different controlled tasks.

**Weaknesses:**

I want to call this sections "Potential Improvements" rather than weaknesses. And this is the part where I want to say most of the stuff I think needed to be discussed.

### **1) Dataset maturity and consistency:**
The sample provided with the paper includes 100 examples per task. “TopologicalCounting” files include labels in their names (e.g., example_001_inferno_heatmap), while “TransformationRecognition” uses random UUIDs. Plus "“TopologicalCounting” " comes with a detailed config.json file with extra metadata which is not present on “TopologicalCounting” at all. This inconsistency makes it look like the two were made by two different teams and just put together inside a single file in a rush. A unified structure would make the benchmark easier to use and trust.

### **2) Go beyond zero-shot results:**
Right now all results are zero-shot evaluations. Adding fine-tuned and trained results (both for MLLMs and for pure vision models) would be extremely informative. Because the tasks are text-independent and have fixed answer spaces (5 multiple choice), this setup is perfect for training vision models to compare directly with text based multimodal models. It would also show whether current models’ limitations are about reasoning or simply training mismatch.

### **3) Expand task diversity:**
The core idea is very good. Only two tasks are included, but the idea could support many. You could add symmetry classification, noise robustness (I mean mathematical high frequency noise, not image noise), or even better, parameter estimation tasks. These would highlight the full flexibility of your idea, but right now the code idea is just undersold with two simple tasks.

### **4) Include in-depth cross-representation analysis:**
While reading the paper, I noticed an interesting connection with another recent work. Your emphasis on representation, specially use of different "Plot Types" mentioned on A.2, and the conclusion you make: "The data reveals a clear performance drop for most models when plot styles differ" is the core idea of the VGA paper (vga.csail.mit.edu) which they call it "conceptualization". Your paper is the third I’ve seen to clearly demonstrate this representation-sensitivity problem in LLMs. It would be very informative to include a deeper analysis of how different visual representations affect model performance. An interesting extension would be to replicate VGA’s core experiment in your functional plots: test whether models can recognize that two plots represent the same function when rendered in different visual styles.

### **Minors:**
#### **5) Scale up to make a dataset, not only a benchmark:** Since the data are synthetic it should be easy to generate hundreds of thousands of examples. A dataset let us train and fine-tune models for deeper analysis.
#### **6) Structure and define  `config.json` :** It's not very clear what to do with `config.json`. It always has 4 `null` values. I really couldn't make sense what is `func_id` (which is always `null` for the second function). This should be clearly document in your paper.

### **Final Remarks**
I want to emphasize that I really like the idea behind this paper. The current paper feel unfinished, and it can be much better. This could become a strong and influential benchmark if enough effort were to be put into it. My recommendation to reject is purely about the current state, not the idea. I want to see this paper at it's mature point then I definitely vote for acceptance.

**Questions:**

I think the paper is clear and I don't have much questions. only one.

About “OCR-proof” you said "xes/labels are held constant across pairs, so models must reason
about the surface itself rather than read values". But that's not the case for the actual numbers shown in the plots. Don't you think models can be biased toward certain numbers? In other words, if you mask the values would the results be consistant?

---

> ### Author Response · Authors · 2025-11-26
> **Response to reviewer FBeL - Part 1/3**
>
> We thank the reviewer for recognizing the clarity and extensibility of our benchmark design, and for valuing our diagnostic analysis that distinguishes between genuine reasoning failures and heuristic shortcuts.
>
> ### 1) Dataset maturity and consistency
>
> > **Reviewer Comment:**
> > The sample provided with the paper includes 100 examples per task. “TopologicalCounting” files include labels in their names (e.g., example_001_inferno_heatmap), while “TransformationRecognition” uses random UUIDs. Plus “TopologicalCounting” comes with a detailed config.json file with extra metadata which is not present on “TopologicalCounting” at all. This inconsistency makes it look like the two were made by two different teams and just put together inside a single file in a rush. A unified structure would make the benchmark easier to use and trust.
> >
> > **Structure and define config.json :** It's not very clear what to do with config.json. It always has 4 null values. I really couldn't make sense what is func_id (which is always null for the second function). This should be clearly document in your paper.
>
> We thank the reviewer for pointing out the lack of clarity in the data structure. We have updated the supplementary material to include a cleaned and unified format for the data dictionary. We removed the unnecessary redundant fields to ensure the data is transparent and easier to use. We will ensure that the final released version of the MaRVL-QA dataset adheres to this new, unified data format.
>
> ---
>
> ### 2) Go beyond zero-shot results
>
> > **Reviewer Comment:**
> > Right now all results are zero-shot evaluations. Adding fine-tuned and trained results (both for MLLMs and for pure vision models) would be extremely informative. Because the tasks are text-independent and have fixed answer spaces (5 multiple choice), this setup is perfect for training vision models to compare directly with text based multimodal models. It would also show whether current models’ limitations are about reasoning or simply training mismatch.
>
> We thank the reviewer for this constructive and critical suggestion. We agree that including trained results for a non-multimodal vision model provides a necessary baseline for a comprehensive interpretation of the zero-shot performance of Multimodal Large Language Models (MLLMs).
>
> **Action Taken: CV Baseline Implementation**
> We trained a Computer Vision baseline using ResNet50 (initialized using pre-trained weights) on the Transformation Recognition task. This established a performance ceiling for a model focused purely on visual classification, allowing us to isolate the MLLMs' zero-shot reasoning deficiencies.
>
> **Model and Training Details:**
> * **Model:** ResNet50 Backbone (~24 million parameters).
> * **Dataset Split (Zero Function Overlap):**
>     * **Training Set:** 58,725 examples from 279 unique functions.
>     * **Testing Set:** 388 examples from 38 separate, held-out unique functions. The test set was strategically sampled to ensure representation across all plot types and applicable transformations.
> * **Training Hyperparameters:** Trained for 50 epochs (with loss converging in ~10 epochs), using a batch size of 128, a learning rate of 1e-3 and Adam optimizer with ReduceLROnPlateau scheduler.
>
> **Comparative Results:**
>
> | Model Type | Model Name | Training Paradigm | Accuracy (%) |
> | :--- | :--- | :--- | :--- |
> | **Pure Vision** | **ResNet50** | **Supervised Training** | **87.11** |
> | Multimodal | O4-mini | Zero-Shot | 57.73 |
> | Multimodal | O3 | Zero-Shot | 49.74 |
> | Multimodal | Sonnet-4 | Zero-Shot | 24.74 |
>
> **Justification: Strengthening the MaRVL-QA Baseline**
> The large gap between the fine-tuned ResNet50 and the zero-shot MLLMs is highly informative and directly strengthens our paper's core argument:
>
> 1.  **Confirmed Task Solvability**
>     The high accuracy of the supervised ResNet50 (87.11%) establishes a visual feasibility oracle, proving the task is visually solvable and eliminating the possibility that MLLM failures are due to visually ambiguous data or inherent task difficulty.
> 2.  **Isolating the Reasoning Gap**
>     The MLLMs' zero-shot failure is, therefore, not a simple "training mismatch" for visual classification. If the MLLMs' vision modules were merely sub-optimal, their performance would still be expected to be significantly higher given the simplicity of the visual task and their vast pre-training. Instead, the persistent performance gap points to a failure to perform zero-shot generalization. The MLLMs fail to successfully:
>     * Generalize abstract mathematical concepts (their symbolic knowledge) to a novel visual domain (the plots).
>     * Couple their visual observation with the correct abstract textual label.
>
> This confirms that MaRVL-QA successfully isolates a genuine mathematical-visual reasoning bottleneck and is a valuable benchmark for testing abstract knowledge transfer in MLLMs.
>
> We will update the paper to include these results and the corresponding discussion.
>
> ---

---

> > ### Author Response · Authors · 2025-11-26
> > **Response to reviewer FBeL - Part 2/3**
> >
> > ### 3) Expand task diversity
> >
> > > **Reviewer Comment:**
> > > The core idea is very good. Only two tasks are included, but the idea could support many. You could add symmetry classification, noise robustness (I mean mathematical high frequency noise, not image noise), or even better, parameter estimation tasks. These would highlight the full flexibility of your idea, but right now the code idea is just undersold with two simple tasks.
> >
> > Thank you for the encouraging feedback on the core idea of MaRVL-QA and for recognizing the flexibility of our proposed framework. We agree that the engine behind MaRVL-QA can support a wide variety of tasks.
> > However, our decision to focus on Topological Counting and Transformation Recognition and to exclude tasks like Symmetry Classification and Parameter Estimation was a deliberate choice to maintain mathematical rigor and avoid "visual ill-posedness."
> >
> > We address your specific suggestions below:
> >
> > **1. Why we excluded Symmetry and Parameter Estimation (The Finite Window Problem)**
> > You suggested tasks like Symmetry Classification or Parameter Estimation. While valuable in a symbolic context, these become mathematically ambiguous when restricted to a "visual landscape" (a finite plot window).
> > * **Ambiguity:** A function like $f(x) = \sin(x)$ plotted over $[0, 2\pi]$ appears to have perfect rotational symmetry. However, without external symbolic knowledge, the behavior outside this window is unconstrained (e.g., the function could be defined as $f(x)=0$ for $x > 2\pi$).
> > * **Hallucination Risk:** Asking a model to infer global properties (symmetry) or inverse parameters from a finite crop forces the model to hallucinate information not present in the pixels. We established a "Rigorous Ambiguity Filtering" pipeline to ensure that every ground truth is objectively verifiable solely from the visual evidence provided.
> >
> > **2. Visual Reasoning vs. OCR/Arithmetic**
> > Parameter estimation often devolves into an "OCR + Arithmetic" task (reading axis ticks and calculating slopes) rather than pure visual reasoning. Our goal was to test the model's ability to reason about spatial relationships and topological features (maxima/minima) in the absence of explicit semantic cues, rather than its ability to read numbers.
> >
> > ---
> >
> > ### About “OCR-proof”
> >
> > > **Reviewer Comment:**
> > > you said "xes/labels are held constant across pairs, so models must reason about the surface itself rather than read values". But that's not the case for the actual numbers shown in the plots. Don't you think models can be biased toward certain numbers? In other words, if you mask the values would the results be consistent?
> >
> > The reviewer raises an important methodological concern regarding the potential for numeric axis labels to introduce an OCR-based heuristic, undermining our claim that the Transformation Recognition task requires visual-geometric reasoning.
> > To address this, we conducted an ablation study where the numeric axis labels were removed. We evaluated five representative models on a 200-item subset, yielding the following results:
> >
> > | Model | Baseline Acc. (%) | No Axis Numbers Acc. (%) | Δ Acc. (pp) |
> > | :--- | :--- | :--- | :--- |
> > | OpenAI o4-mini | 65.0 | 56.50 | -8.50 |
> > | OpenAI o3 | 60.0 | 54.00 | -6.00 |
> > | Claude Sonnet 4 | 33.5 | 33.00 | -0.50 |
> > | Pixtral Large | 18.0 | 17.00 | -1.00 |
> > | Mistral Medium | 17.0 | 17.00 | 0.00 |
> >
> > **Analysis:**
> > * **Brittle and Heuristic Model Insensitivity:** For the models categorized as Brittle Reasoners (Claude Sonnet 4 ) and Disengaged Heuristics (Pixtral Large, Mistral Medium), the change in performance is negligible ($\Delta \le 1.0$ pp). This insensitivity to the presence of explicit numerical grounding confirms that these models do not systematically rely on the axis number values to solve the transformation task, aligning with the paper's conclusion that they use fragile or non-visual strategies.
> > * **Top-Tier Model Sensitivity to Increased Difficulty:** For the top-performing models, OpenAI o4-mini and OpenAI o3, a moderate performance drop is observed ($\Delta=-8.5$ pp and $\Delta=-6.0$ pp). Crucially, their post-ablation accuracy remains significantly high (well above 50%), confirming a core reliance on visual-geometric reasoning. The observed drop is likely attributed to the increase in the intrinsic difficulty of the task. By removing the explicit coordinate values, the models lose the numerical grounding required to infer the precise scale and magnitude of the transformation, forcing them to rely exclusively on the abstract visual pattern, which is a demonstrably harder inferential task.

---

> > > ### Author Response · Authors · 2025-11-26
> > > **Response to reviewer FBeL - Part 3/3**
> > >
> > > ### 4) Include in-depth cross-representation analysis
> > >
> > > > **Reviewer Comment:**
> > > > While reading the paper, I noticed an interesting connection with another recent work. Your emphasis on representation, specially use of different "Plot Types" mentioned on A.2, and the conclusion you make: "The data reveals a clear performance drop for most models when plot styles differ" is the core idea of the VGA paper (vga.csail.mit.edu) which they call it "conceptualization". Your paper is the third I’ve seen to clearly demonstrate this representation-sensitivity problem in LLMs. It would be very informative to include a deeper analysis of how different visual representations affect model performance. An interesting extension would be to replicate VGA’s core experiment in your functional plots: test whether models can recognize that two plots represent the same function when rendered in different visual styles.
> > >
> > > We sincerely thank the reviewer for highlighting the compelling connection between our findings on representation sensitivity and the "conceptualization" problem documented in recent literature, such as the VGA paper.
> > >
> > > **New Experiment: Functional Equivalence Recognition**
> > > To directly address this point and isolate the visual abstraction capacity from the complexity of geometric manipulation, we conducted a new Functional Equivalence Recognition experiment, conceptually replicating the spirit of the VGA benchmark within our controlled functional landscape.
> > >
> > > **Task:** Given two plots (A and B), identify if they represent the same underlying mathematical function, regardless of visual rendering style.
> > > The experiment used 300 QA pairs, balanced between plots with the same style and plots rendered in different styles. These samples were generated with representation from all function families, ensuring the same rigorous diversity as the stratified sampling strategy used for our main Transformation Recognition task.
> > >
> > > | Model | Functional Equivalence Recognition (Accuracy %) |
> > > | :--- | :--- |
> > > | OpenAI o3 | 94.00% |
> > > | OpenAI o4-mini | 90.67% |
> > > | Qwen-2.5 VL Max | 90.67% |
> > > | Pixtral Large | 88.67% |
> > > | Claude Sonnet 4 | 84.33% |
> > > | Mistral Medium | 79.00% |
> > >
> > > **Analysis and Integration of New Finding**
> > > This new data provides a crucial nuance to our overall narrative, which we will integrate into the paper's discussion:
> > > * **Distinction Between Abstraction and Procedural Reasoning:** The results show that MLLMs are highly proficient (scoring between 79% and 94%) at the core task of style-agnostic functional abstraction (recognizing equivalent functions). This capacity requires successfully factoring out superficial visual changes (e.g., color map, plot type) to match the underlying topological structure of the function $z=f(x,y)$.
> > > * **A New Bottleneck:** When models fail Transformation Recognition, it is not because they cannot recognize the underlying function (as demonstrated by the high equivalence scores), but because they cannot accurately map the spatial change between two abstract representations. This pins the failure on the procedural geometric manipulation component (rotations/translations) rather than a mere failure in visual conceptualization or feature extraction.
> > >
> > > This extended analysis strengthens our paper by demonstrating that while MLLMs possess a strong, style-agnostic ability for functional conceptualization, this capacity does not translate into a robust ability for systematic spatial and geometric reasoning. We will include this new experiment and analysis in a dedicated subsection in the final manuscript.
> > >
> > > ---
> > >
> > > ### 5) Scale up to make a dataset, not only a benchmark
> > >
> > > > **Reviewer Comment:**
> > > > Since the data are synthetic it should be easy to generate hundreds of thousands of examples. A dataset let us train and fine-tune models for deeper analysis.
> > >
> > > We fully agree with the reviewer that providing a large-scale, synthetic dataset for model fine-tuning is an important contribution for the community.
> > >
> > > For the Transformation Recognition task, the collection of 79,542 high-confidence QA pairs generated by our pipeline for the Transformation Recognition task can be utilized as a large-scale training and test split. This volume is suitable for the training and analysis of foundational models, a claim supported by the training of our ResNet 50 baseline model on this data (results in point 2). In our next version, we plan to further increase our dataset size from ~80k to ~500k.
> > >
> > > For the Topological Counting task, the size constraint is a consequence of our commitment to high quality objective ground truth. As detailed in Section 3.2.1, generating the final 1,548 QA pairs required meticulous manual curation by the authors to certify that the programmatically identified count exactly matched the number of clearly visible features. The requirement for this manual, human-verified certification step means that large-scale generation for training purposes is not the intended scope for this specific task.

---

### Author Response · Authors · 2025-11-26
**Summary of New Experiments and Revisions**

We thank the reviewers (FBeL, gLLJ, kggk, azUV) for their constructive feedback and for highlighting the benchmark's "very good core idea" and "brilliant analysis" of model failure modes.
In response to your collective requests, we have conducted extensive new experiments and analyses. Below is a summary of the major updates included in our revision/rebuttal.

### 1. Expanded Model Evaluation (Addressing Reviewer gLLJ)
To ensure our claims of failure apply to modern architectures, we expanded our evaluation beyond the LLaVA family to include 5 state-of-the-art open-source models: Qwen 3 (8B & 235B), Gemma 3 (12B, 27B), and Phi 4 1.
**Result:** The "catastrophic breakdown" in enumeration persists. Even the massive Qwen 3 235B achieves only 44.6% on counting, significantly trailing proprietary models. This demonstrates that the deficit is not an artifact of older models, but persists even in the newest open-source architectures.

### 2. New Baselines and Ablations (Addressing Reviewers FBeL, kggk)
* **ResNet50 Supervised Baseline:** We trained a ResNet50 vision baseline on the Transformation task (Reviewer FBeL). It achieved 87.11% accuracy (vs. 57.7% for o4-mini) on a newly sampled test set. This establishes a "visual feasibility oracle", demonstrating that the task is visually solvable and implying that MLLM failures stem from reasoning deficits rather than visual ambiguity.
* **Numerical Label Ablation:** We re-evaluated models with axis numbers removed (Reviewers FBeL, kggk). Top models (o4-mini, o3) saw a moderate drop (~6-8%) but remained capable, indicating that explicit numerical cues are not required for their performance. For models like Pixtral and Mistral, performance remained identical. This suggests they do not process the axis values, but instead default to fixed guessing strategies regardless of the visual input.
* **Grid Lines:** Adding grid lines yielded no significant change in performance ($<2\%$ variation) across all models, confirming that the bottleneck is reasoning, not visual alignment.

### 3. Diagnostic Analysis (Addressing Reviewers FBeL, gLLJ, azUV)
* **Functional Equivalence (The "VGA" Experiment):** We tested if models can recognize the same function across different styles (Reviewer FBeL). Models scored highly (90%+), yet failed to identify transformations of those same functions. This suggests the bottleneck lies in spatial reasoning (manipulating the plot), not in perception (recognizing the underlying function).
* **Statistical Significance:** We applied McNemar’s tests (Reviewer azUV) to confirm that performance differences were real. The results show that while the top model (o4-mini) is significantly better than others at Transformations, the top models are statistically indistinguishable from each other on Counting. This indicates that the inability to enumerate features is a common limitation across these systems, rather than a model-specific fluctuation.
* **Complexity Analysis:** We broke down performance by surface topology (Reviewer azUV). Accuracy collapses on complex surfaces (e.g., Composite/Modulated) compared to simple Periodic surfaces. This confirms that the current counting capabilities of MLLMs are brittle and fail to generalize to irregular or non-repetitive topologies.

### 4. Data Hygiene and Standardization (Addressing Reviewer FBeL)
We have cleaned the dataset structure, unifying the JSON schema across tasks and removing redundant fields to ensure the benchmark is user-friendly. We have re-uploaded the updated data in the supplementary material.

We believe these additions significantly strengthen the paper and validate MaRVL-QA as a critical diagnostic tool for the MLLM community. We have provided detailed point-by-point responses to each reviewer below. Given the clarifications and additional experiments, we hope that reviewers reconsider the ratings.

---

### Author Response · Authors · 2025-12-04
**Summary of Rebuttal Updates for AC**

**To the Area Chair**,

We understand that due to the recent administrative reset, the visible reviews and scores reflect the submission in its initial state, prior to the extensive experiments and updates conducted during the discussion period.

We specifically highlight that Reviewer FBeL (Score: 2) explicitly stated: "*My recommendation to reject is purely about the current state... I want to see this paper at its mature point then I definitely vote for acceptance.*"

To assist you in your evaluation, we have prepared this roadmap showing how the revised manuscript fulfills the specific criteria for **maturity** and **completeness** requested by the reviewers:

**1. Dataset Maturity & Consistency (Addressing Reviewer FBeL)**
- **Critique:** The initial data release contained inconsistent file structures and naming conventions between tasks.
- **Resolution:** We have fully standardized the JSON schema and file naming across both tasks. The dataset is now unified, clean, and user-friendly, and the updated data dictionary has been uploaded to the supplementary material.

**2. Baselines & Open-Source Diversity (Addressing Reviewers FBeL, gLLJ)**
- **Critique:** Reviewers requested a pure vision baseline to prove task solvability and a broader evaluation of modern open-source models beyond the LLaVA family.
- **Resolution (New Experiments):**
    - **Vision Baseline:** We trained a supervised **ResNet50** on the Transformation task, achieving **87.11%** accuracy (vs. the best MLLM, o4-mini, at **57.73%**). This confirms the task is visually solvable and isolates the MLLM failure as a reasoning deficit.
    - **Modern Open-Source:** We expanded our evaluation to include **Qwen 3 (8B & 235B), Gemma 3 (12B & 27B), and Phi 4**. The results confirm that foundational failures persist even in the newest architectures (e.g., Qwen 3 235B at 44.6% on counting).

**3. Task Diversity & "Visual Conceptualization" (Addressing Reviewer FBeL)**
- **Critique:** The reviewer suggested testing if models can recognize the same function across styles to isolate visual perception from geometric reasoning.
- **Resolution (New Experiment):** We implemented a **Functional Equivalence Recognition** task. Models scored highly (**90%+**) on identifying different representations of the same underlying function, but failed to identify transformations of those same functions. This pinpoints the bottleneck: models possess "visual conceptualization" but lack procedural geometric reasoning.

**4. Robustness & Ablations (Addressing Reviewers kggk, azUV)**
- **Critique:** Reviewers requested analysis on OCR reliance (axis numbers), grid lines, and statistical significance.
- **Resolution (New Analysis):**
    - **Ablations:** We conducted two ablations to isolate failure modes. **Adding Grid Lines** provided no performance boost, ruling out visual alignment issues. **Removing Axis Numbers** caused a moderate drop for top models but **0% change** for other models.
    - **Statistical Significance:** We applied **McNemar’s** tests, which revealed that while o4-mini outperforms others in Transformations, the top models are statistically indistinguishable in Counting, pointing to a shared architectural ceiling.

**Conclusion**

We believe the addition of the **ResNet50 baseline, 5 new open-source models, the functional equivalence experiment, and the cleaned dataset** directly addresses the concerns found in the initial reviews.
We hope this roadmap assists you in assessing the paper's revised standing.

---

### Note · Program_Chairs · 2026-01-17
**Submission Desk Rejected by Program Chairs**

The following references in this submission do not refer to real documents and/or have major errors in bibliographic information:

 Zu-Yu Liu, Zhaida Wang, Matt Gwilliam, Jinyue Cong, Jiacheng Li, He Ji, Ting-Hao Chang, and Heng Yu. Evaluating compositional structures in text-to-image models. arXiv preprint arXiv:2311.10773, 2023c.
Fangyu Liu, Julian Mintz, Shuo Cheng, Linyong Hou, Chao Zhang, Da Chen, and Jing Li. DePlot: One-shot visual language localization. arXiv preprint arXiv:2303.04780, 2023a.
Nitesh Methani, Pratik Gpt, M. El-allaui, V. Varma, and K. Sankaranarayanan. PlotQA: Reasoning over scientific plots. In Proceedings of the 2020 Conference on Empirical Methods in Natural Language Processing: Findings, pp. 417-422, 2020.
Jiahao Chen, Shuo Cheng, Linyong Hou, Chao Zhang, and Jing Li. UniGeo: A unified framework for solving geometry problems. arXiv preprint arXiv:2311.18430, 2023.